# Targeting the conserved active site of splicing machines with specific and selective small molecule modulators

Ilaria Silvestri [1,2,5], Jacopo Manigrasso[1,6], Alessandro Andreani[1], Nicoletta Brindani [1], Caroline Mas [3], Jean-Baptiste Reiser [4], Pietro Vidossich[1], Gianfranco Martino[1], Andrew A. McCarthy [2], Marco De Vivo [1] ✉ & Marco Marcia [2] ✉

The self-splicing group II introns are bacterial and organellar ancestors of the nuclear spliceosome and retro-transposable elements of pharmacological and biotechnological importance. Integrating enzymatic, crystallographic, and simulation studies, we demonstrate how these introns recognize small molecules through their conserved active site. These RNA-binding small molecules selectively inhibit the two steps of splicing by adopting distinctive poses at different stages of catalysis, and by preventing crucial active site conformational changes that are essential for splicing progression. Our data exemplify the enormous power of RNA binders to mechanistically probe vital cellular pathways. Most importantly, by proving that the evolutionarily-conserved RNA core of splicing machines can recognize small molecules specifically, our work provides a solid basis for the rational design of splicing modulators not only against bacterial and organellar introns, but also against the human spliceosome, which is a validated drug target for the treatment of congenital diseases and cancers.

Splicing is a ubiquitous and essential biological reaction whereby protein-coding or regulatory RNAs (the exons) are excised from precursor transcripts by removing intronic sequences. In prokaryotes and eukaryotic organelles, splicing is performed by auto-catalytic introns, some of which regulate the expression of vital metabolic genes[1]. Of these, the so-called group II introns constitute the ancestors of the eukaryotic spliceosome, a megadalton-large ribonucleoprotein complex that ensures the correct maturation of ~90% of human genes and whose aberrant activity causes ~15% of all human hereditary diseases and cancers[2]. Besides catalyzing their self-splicing, group II introns also act as mobile retroelements, thus crucially contributing to genetic diversity throughout evolution[3]. Importantly, as splicing ribozymes

and retroelements, group II introns are potential targets of antifungal agents and molecular machines that can be engineered for site-specific insertion of cargo genes into genomic DNA[1]. Therefore, their modulation with small RNA-targeting molecules has enormous potential to probe fundamental biological mechanisms, with direct translational applications in biotechnology and human medicine[4].

Importantly, small organic compounds have recently been shown to target nuclear and organellar splicing complexes[5,6]. Such breakthroughs led to the approval of the drug risdiplam for the treatment of neuromuscular disorders, and to the design of the lead compound intronistat B for inhibiting the growth of pathogenic fungi[5,6]. However, the mode of action and mechanism of these few compounds remain

[1]Laboratory of Molecular Modelling & Drug Discovery, Istituto Italiano di Tecnologia, Via Morego 30, 16163 Genoa, Italy. [2]European Molecular Biology Laboratory (EMBL) Grenoble, 71 Avenue des Martyrs, Grenoble 38042, France. [3]Univ. Grenoble Alpes, CNRS, CEA, EMBL, ISBG, F-38000 Grenoble, France. [4]Univ. Grenoble Alpes, CNRS, CEA, IBS, F-38000 Grenoble, France. [5]Present address: Institute of Crystallography, National Research Council, Via Vivaldi 43, 81100 Caserta, Italy. [6]Present address: Medicinal Chemistry, Research and Early Development, Cardiovascular, Renal and Metabolism (CVRM), BioPharmaceuticals R&D, AstraZeneca, Gothenburg, Sweden. ✉e-mail: marco.devivo@iit.it; mmarcia@embl.fr

uncertain, mainly due to the lack of structural information about the RNA-ligand complex.

As a matter of fact, with only 841 structures of RNA-ligand complexes in the Protein Data Bank (PDB) as of January 16th, 2024, the exact comprehension of how small molecules specifically and/or selectively bind RNA remains quite challenging[7,8]. RNA-binding molecules typically tend to be unspecific and promiscuous, by interacting in a sequence-independent manner with the phosphodiester backbone of RNA or by intercalating stacked nucleobases inducing structural misfolding[4].

In this context, we and others have previously demonstrated that both the eukaryotic spliceosome and the bacterial group II introns possess an evolutionarily and structurally-conserved active site[9–11]. This RNA catalytic core binds a cluster of di- and mono-valent metal ions, which acts as an indispensable cofactor for splicing[9–12]. Both splicing machines follow the same reaction chemistry, which consists of two steps. During the first step of splicing, the 5′-exon is excised through a nucleophilic reaction that induces a transiently-inactive 'toggled' state, necessary to recruit the reactants of the second step[9,13,14]. Then, in the second step of splicing, the free 5′-exon attacks the 3′-splice junction, producing ligated exons and a free intron[1]. This vital and multistep process involves sequence-specific interactions within a highly-structured metal-aided catalytic site, which in principle offers unique opportunities of favorable RNA-ligand interactions.

Our work now shows how the conserved RNA-based active site of splicing ribozymes can indeed recognize small organic compounds. These compounds anchor to catalytic nucleotides and metal ions, compete with the splice junctions, and selectively prevent active site structural rearrangements required for splicing progression. Our work thus reveals that splicing modulation can occur through direct targeting of the splicing site. These unforeseen findings, obtained through the integration of enzymatic, computational, and crystallographic studies, determine the exact mechanism of inhibition of bacterial and organellar splicing, with important implications for the design of antifungal antibiotics. More broadly, they also provide the basis for future design of spliceosomal active site modulators that establish nucleotide-specific interactions around the splice junctions. Such a targeting approach, which was impossible to practically envision until now, has great potential for the treatment of congenital neurological disorders and cancers that derive from alternative splicing defects.

## Results

### Bacterial and organellar splicing is inhibited by small molecules in a step-specific manner

Following the recent identification of small molecules as chemical probes of mitochondrial group IIB introns[5], here we set out to elucidate their molecular mechanism and to explore whether these compounds could be used as tools to gain mechanistic insights into the different steps of the splicing reaction, an approach that had been possible until now only through conventional mutagenesis of active site residues or through replacement of active site metal ions[9–11].

We first tested whether these molecules inhibit bacterial group IIC introns, besides their mitochondrial homologs, because the I1 intron from *Oceanobacillus (O.) iheyensis* (secondary structure map reported in Supplementary Fig. S1) can be readily crystallized and would thus enable us to obtain mechanistic insights at high-resolution.

Using an established radio-analytic self-splicing assay, we determined that the most active compound, i.e. intronistat B, inhibits *O. iheyensis* group IIC intron with a $K_i = 1.700 \pm 0.004 \, \mu M$ (Fig. 1 and Supplementary Fig. S2, Supplementary Table S1, see chemical structure of intronistat B in Fig. 5). For reference, intronistat B inhibits the ai5γ group IIB intron from *Saccharomyces (S.) cerevisiae* with $K_i = 0.360 \pm 0.020 \, \mu M$[5]. Of note, an orthogonal FRET assay, which probes the multiple-turnover spliced-exon reopening (SER) reaction

also catalyzed by group II introns, allowed us to confirm an $IC_{50}$ value of $2.5 \pm 0.7 \, \mu M$ for the ai5γ group IIB intron, similar to previous reports[5]. Unfortunately, this assay did not work for the *O. iheyensis* group IIC intron, possibly owing to the fact that the shorter exon-binding site sequence (EBS1) in this intron makes the SER reaction less efficient[15]. We have thus estimated an apparent $IC_{50}$ of $2.3 \pm 0.4 \, \mu M$ for the *O. iheyensis* group IIC intron, as calculated from the radio-analytic self-splicing assay at 15 min post-addition of the compound. Importantly, we further established that intronistat B targets the folded intron in its active state. This observation is demonstrated by the inhibitory effect of intronistat B, which is evident also when the compound is added to an ongoing splicing reaction (Supplementary Fig. S3). Direct binding of intronistat B to the intron was also assed using both bio-layer interferometry (BLI) and isothermal titration calorimetry (ITC). Both methods confirmed an interaction with micromolar affinity ($K_{D, ITC} = 7.58 \pm 1.53 \, \mu M$ and $K_{D1, BLI} = 128 \pm 12 \, \mu M$ and $K_{D2, BLI} = 201 \pm 11 \, \mu M$) corresponding to fast kinetics rates as determined with BLI ($k_{on1} = 10.1 \pm 0.9 \, M^{-1} \cdot s^{-1}$, $k_{on2} = 132.0 \pm 6.7 \, M^{-1} \cdot s^{-1}$, $k_{off1} = 0.001 \pm 0.000 \, s^{-1}$, $k_{off2} = 0.026 \pm 0.000 \, s^{-1}$) and through an enthalpy-driven reaction as shown by ITC ($\Delta H = -10.04 \pm 0.89 \, kcal/mol$, $\Delta G = -7.02 \pm 0.14 \, kcal/mol$, $-T\Delta S = 3.03 \pm 0.76 \, kcal/mol$) (Supplementary Figs. S4 and S5). Besides informing on the overall potency of the compound, our assay allowed us to discriminate the effects of the compound on the first vs the second steps of splicing, because the *O. iheyensis* group IIC intron produces linear intron-3′-exon intermediates (I-3E) and free introns (I) that can be resolved by electrophoresis. Our data proved that in the presence of intermediate intronistat B concentrations (10-50 μM), the first step of splicing is ~5-fold slower than in the absence of the compound ($k_1 = 0.037 \pm 0.002 \, min^{-1}$ without compound, $k_1 = 0.007 \pm 0.002 \, min^{-1}$ at 50 μM intronistat B, Supplementary Table S1). Unexpectedly, we additionally noticed that intronistat B induces accumulation of I-3E, suggesting that the intron is unable to progress on to the second step of splicing. Indeed, our kinetic analysis proved that the second step of splicing is 34-fold slower in the presence vs absence of the compound ($k_2 = 0.031 \pm 0.003 \, min^{-1}$ without compound, $k_2 = 0.001 \pm 0.000 \, min^{-1}$ at 50 μM intronistat B, Supplementary Table S1). These data indicate that intronistat B selectively inhibits the second step of splicing 7-fold more potently than the first step (Fig. 1 and Supplementary Fig. S2). Importantly, the splicing defects induced by intronistat B are comparable to those caused by active site mutations designed to impair the transition from the first to the second step of splicing, i.e. the so-called G- and U-triple mutants that inhibit active site protonation or the C377G mutant which impair active site toggling[9,13].

In summary, our enzymatic studies establish that intronistat B displays a broader spectrum than previously suggested. It inhibits bacterial group IIC introns, which splice through a hydrolytic mechanism and produce linear introns, with a similar $IC_{50}$ and a 5-fold higher $K_i$ than fungal group IIB introns, which splice through a trans-esterification mechanism producing lariat introns. Moreover, our results show that intronistat B inhibits both the first and the second steps of group II intron splicing with a stronger effect on the second step. This unforeseen discovery could mean that intronistat B binds at or near the splice site with unexpected selectivity.

### Principles of molecular recognition between the conserved group II intron active site and splicing modulators

To visualize the exact binding mode and characterize the inhibitory mechanism of intronistat B at high resolution, we first solved the co-crystal structure of intronistat B bound to a previously-described construct of the *O. iheyensis* I1 intron[9], which encompasses its structural domains 1–5 (OiD1-5), in the exon-free state and in the presence of $Mg^{2+}$ and $K^+$ ions.

Our structure, determined at a resolution of 3.0 Å, shows that in the presence of the compound, the intron maintains an overall folded

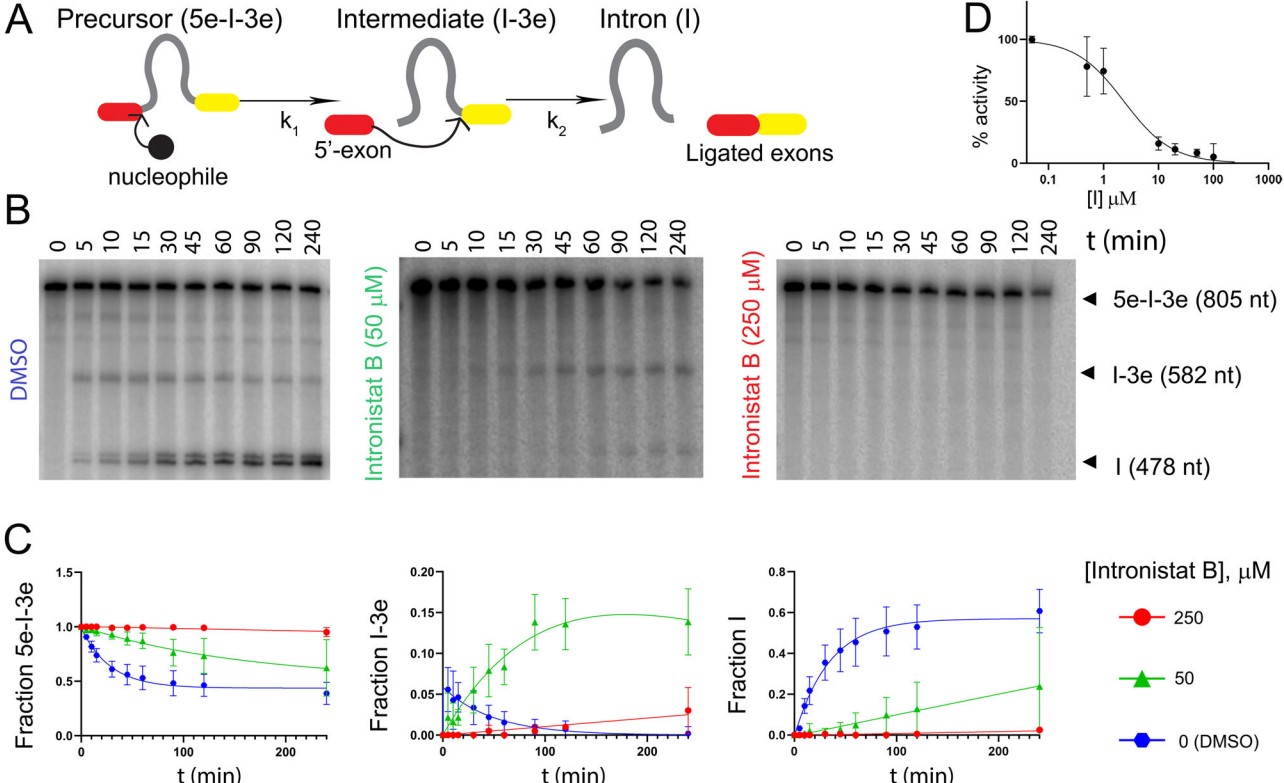

**Fig. 1 | Kinetics of splicing inhibition. A** Schematic of the splicing reaction. $k_1$ is the rate constant of the first and $k_2$ of the second step of splicing. Kinetic rate constants relevant to our study are reported in Supplementary Table S1. **B** Representative splicing kinetics in the absence (control, DMSO) and in the presence of intronistat B (50 and 250 μM). Precursors are indicated as 5e-I-3e (nt length in parenthesis). Intermediate (I-3e) and linear intron (I) migrate as double bands because of cryptic cleavage sites, as explained previously[9,13]. **C** Evolution of the populations of precursor (5e-I-3e, left panel), intermediate (I-3e, middle panel), and linear intron (I, right panel) over time. Error bars represent standard errors of the mean (s.e.m.) calculated from $n = 3$ independent experiments. Complete kinetics including all tested intronistat B concentrations are reported in Supplementary Fig. S2. **D** Dose-response curve in the presence of different concentrations of intronistat B. Error bars represent standard errors of the mean (s.e.m.) calculated from n = 3 independent experiments. Source data are provided as a Source Data file.

structure, similar to that of the compound-free intron (PDB entry 4E8M), with a root-mean-square deviation (RMSD) = 0.5 Å between the two states (Supplementary Table S2). In the intronistat B-bound state the intron active site is intact and adopts the catalytically required triple helix conformation anchored to the M1/M2/K1 metal ion cluster[9]. While the electron density around the metal cluster in the active site is weak, the interatomic distances and coordination spheres were compatible with magnesium ions at the M1 and M2 positions and with potassium ions at the K1 position (Fig. 2B and Supplementary Fig. S28). Importantly, we have modeled magnesium and potassium ions at the M1/M2 and at the K1 positions, respectively, because the identity of the ions that occupy these sites had been previously established based on metal-ion replacement and crystallographic anomalous scattering studies[9,10] (see also a stereo view of the electron density of the active site ions in Supplementary Fig. S28).

To localize the compound in the crystallographic electron density, we calculated the $F_o$-$F_c$ electron density omit-map of the refined structure before modeling the ligand. The map shows an extended positive peak in the active site (highest contour value = 4.5 σ, volume of the peak at 3.0 σ = 449 Å³), compatible with the chemical structure and molecular volume of intronistat B (predicted volume of intronistat B = 333 Å³, Fig. 2). Here, intronistat B makes specific contacts through its pyrogallol group with both M1 (2.0 Å from intronistat B O24 and O26 atoms) and M2 (2.0 Å from intronistat B O22 and O24 atoms, Supplementary Fig. S6A). As shown by our quantum-level calculations, this binding mode is possible, because intronistat B is deprotonated at the hydroxyl group in *para* in the pyrogallol moiety (pK$_A$ = 5.7),

analogously to other similar chelating groups[16]. The pyrogallol moiety makes further hydrogen bond interactions with evolutionarily-conserved active site nucleotides, i.e. the catalytic triad and the 2-nucleotide bulge (Fig. 2B and Supplementary Fig. S6A). Additionally, intronistat B establishes a sequence-specific hydrogen bond with exon binding site 1 (EBS1), the motif that recognizes the exon sequence and thus defines the splicing site (3.0 Å between O18 in intronistat B and N1 in A181, Fig. 2B and Supplementary Fig. S6A). In this conformation, the real-space correlation coefficient (RSCC) calculated for intronistat B after refinement is 0.94, indicating very close similarity between the experimental and theoretical electron-density map for the compound[17].

To further validate this highly-specific compound binding mode, we additionally synthesized a di-brominated intronistat B derivative (ARN25850, Supplementary Figs. S7 and S29). We proved that this compound inhibits both *S. cerevisiae* group IIB (IC$_{50}$ = 6.0 ± 0.9 μM) and *O. iheyensis* group IIC introns (K$_i$ = 48.39 ± 17.09 μM, Supplementary Fig. S7 and Supplementary Table S1). We then solved the crystal structure of OiD1-5 in complex with di-brominated intronistat B at a resolution of 2.8 Å (RMSD = 0.6 with respect to the crystal structure of intronistat B-bound OiD1-5, Supplementary Table S2). The crystallographic data, collected at λ = 0.92 Å, a wavelength where bromide displays an anomalous scattering coefficient *f″* = 4 electrons, enabled us to precisely localize the bromide atoms[18]. The anomalous difference Fourier electron density map derived from our crystallographic data set shows two pronounced peaks within the intron core (highest contour value = 9.2 σ and 11 σ, respectively), at the expected *ortho*

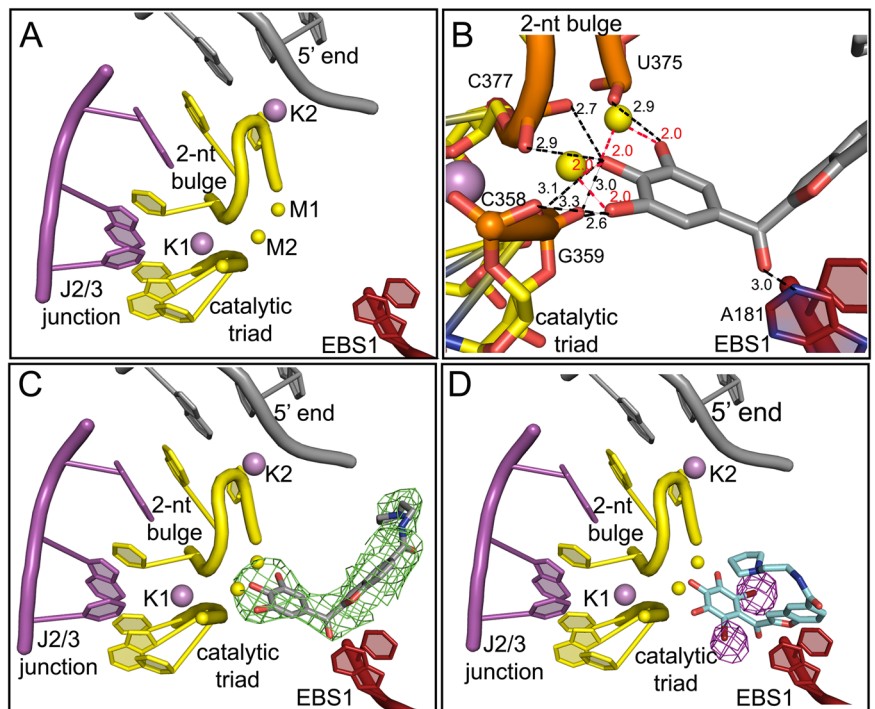

**Fig. 2 | Binding of intronistat B in the exon-free active site of OiD1-5. A** Crystal structure of the apo OiD1-5 in the presence of $Mg^{2+}$ (yellow spheres), and $K^+$ (purple spheres) (PDB entry 4E8M). The main active site elements, i.e. the J2/3 junction (in magenta), the catalytic triad (in yellow), the 2-nucleotide bulge (in yellow) and the EBS1 (in firebrick), are shown as cartoons. **B** Crystal structure of OiD1-5 in the presence of $Mg^{2+}$ (yellow spheres), $K^+$ (purple spheres) and intronistat B (gray sticks) (PDB entry 8OLS). The interactions of intronistat B with the active site elements are shown as black dotted lines (distances in Å, in black for H-bonds and in red for coordination bonds). The representation of intronistat B color-coded by crystallographic B-factor is reported in Supplementary Fig. S27. **C** Same active site as in panel **A** and **B**. Here, the $F_o$-$F_c$ omit map obtained before modeling intronistat B and contoured at 3σ is depicted in green mesh. **D** Crystal structure of OiD1-5 in the presence of $Mg^{2+}$ (yellow spheres), $K^+$ (purple spheres) and the di-brominated intronistat B derivative ARN25850 (cyan sticks, RSCC = 0.88, PDB entry 8OLV). The violet mesh represents the anomalous difference Fourier map contoured at 5σ, revealing the position of the bromide atoms.

positions on the pyrogallol moiety, which binds the active site with the same pose as the intronistat B pyrogallol (Fig. 2D). Notably, the bromine atoms introduce a steric hindrance that is likely responsible for the rotation of the benzofuran bicyclic core towards a conformation orthogonal to the intronistat B benzofuran. This structure thus confirms unequivocally our modeling of the intronistat B conformation at the evolutionarily-conserved group II intron active site.

We further probed the interaction by two independent ~500 ns-long molecular dynamics (MD) simulations of the intronistat B-bound OiD1-5 structure, which show that the inhibitor stably maintains the binding pose captured crystallographically ($RMSD_{RNA} = 5.2 \pm 0.4$ Å; $RMSD_{(intronistat\ B)} = 1.5 \pm 0.1$ Å, Supplementary Fig. S8). In this position, the ligand mimics the splicing substrate (i.e. the scissile nucleotide) and forms a pseudo-Michaelis-Menten complex at the intron active site, stabilizing the M1-M2 internuclear distance at a value compatible with catalysis ($d_{M1-M2} = 4.01 \pm 0.46$ Å; for reference in PDB entry 4E8M $d_{M1-M2} = 3.7$ Å, Fig. 3 and Supplementary Figs. S8, S9). It is worth mentioning that formation of such pseudo-Michaelis-Menten complex is guaranteed by the deprotonation of the *para*-OH of the intronistat B pyrogallol moiety, as confirmed by QM/MM calculations (Supplementary Figs. S24, S25).

Taken together, our OiD1-5 structures with intronistat B and its brominated derivative, and the corresponding MD simulations, demonstrate that intronistat B inhibits splicing because it binds to the active site of group II introns, where it forms direct, specific interactions with conserved catalytic nucleotides and metal ions.

## The mechanism of inhibition of the first step of splicing
Establishing that intronistat B binds at the active site of group II introns and inhibits different catalytic steps with different potency, we sought

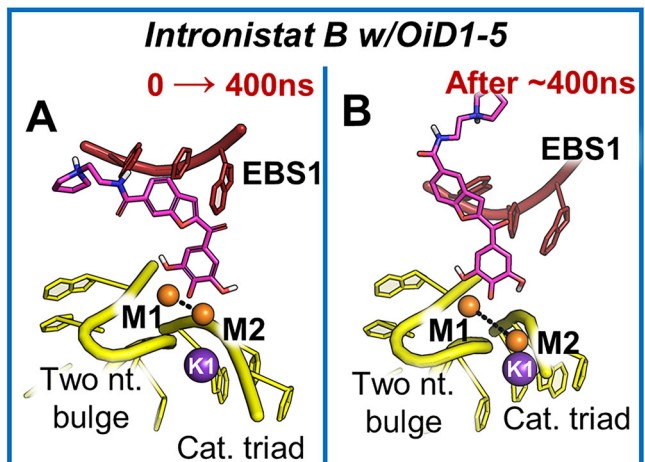

**Fig. 3 | MD simulations. A** Representative binding mode of intronistat B as found during the first 400 ns of MD simulations – when bound to the free intron. The compound (purple sticks) coordinates the two catalytic magnesium ions M1-M2 (orange spheres), thus competing with the splice junction and mimicking a pseudo Michaelis-Menten complex. The catalytic triad (yellow), the two-nucleotide bulge (yellow) and the exon binding site 1 (EBS1, red) residues are shown as cartoon. **B** Representation of the alternative binding mode of intronistat B captured by MD simulations when bound to the free intron. Here, the compound loses the interaction with M2 while preserving the coordination with M1. The representation follows that of panel **A**.

to better understand the underlying mechanism of such unexpected selectivity. To this end, we solved co-crystal structures of *O. iheyensis* group IIC intron in complex with intronistat B at each step of the splicing cycle.

For this study, besides the OiD1-5 construct described above, we additionally used the previously-described Oi5eD1-5 intron, which comprises the short 5′-exon sequence UAUU at its 5′-terminal end[9].

Using Oi5eD1-5, we first obtained a structure of the pre-catalytic state (the stage prior to the first step of splicing) in the presence of $Ca^{2+}$, $K^+$, and intronistat B at 4.0 Å resolution (Supplementary Table S2). The presence of $Ca^{2+}$ inhibits the splicing in group II introns making it possible to visualize the precatalytic state[9]. The simulated-annealing electron density map generated by omitting the G1 at the 5′ splice junction demonstrates that in this structure the active site is occupied only by the reaction substrates, i.e. the 5′-splice junction and the reaction nucleophile. Despite the presence of high concentrations of intronistat B in the crystallization buffer (1 mM, 588-fold over the $K_i$), this structure adopts the exact same conformation reported for the corresponding apo-form (RMSD = 0.6 Å with respect to PDB id: 4FAQ, Supplementary Fig. S10A).

We also determined a co-crystal structure at 3.1 Å resolution of Oi5eD1-5 and intronistat B in the state immediately following the first step of splicing, which is obtained in the presence of $Mg^{2+}$ and $K^+$. Analogously to the structure obtained in the presence of calcium, also the structure obtained in the presence of magnesium shows the compound-free intron, in which the scissile phosphate and the 5′-exon coordinate M1 and M2 in the same conformation reported for the corresponding apo-form (RMSD = 0.4 Å with respect to PDB id: 4FAR, Supplementary Fig. S10B). These two structures of Oi5eD1-5 suggest that under the experimental crystallization conditions (100 mM $Mg^{2+}$, 30 °C) the splice junction outcompetes intronistat B from the active site. We had previously established, however, that immediately after 5′-exon hydrolysis these crystallization conditions trap the scissile phosphate in an unproductive conformation, which is unlikely to be of physiological relevance[13].

To mimic a more physiological conformation of the active site immediately after 5′-exon hydrolysis, we thus co-crystallized OiD1-5 (the free intron) with the 5′-exon-like oligonucleotide 5′-AUUUAU-3′ in the presence of $Mg^{2+}$ and $K^+$, and we solved the X-ray structure of this complex in the absence of inhibitors, and in the presence of intronistat B or the di-brominated intronistat B derivative ARN25850 at different soaking time (1 h or 2.5 h, Supplementary Table S2).

Our OiD1-5 structure without inhibitors reveals a well-folded intron possessing an intact catalytic site that adopts the expected triple helix conformation (RMSD = 0.8 Å with respect to PDB id: 4FAW, Fig. 4A). The 5′-exon-like oligonucleotide added in trans occupies the same position as the spliced oligo ($RMSD_{5'-exon}$ = 0.7 Å with respect to the spliced exon in 4FAW). After 1 h soaking time, intronistat B binds the intron in its active site still coordinating M1 and M2 through its pyrogallol moiety (Fig. 4B). However, in this structure, the $F_o$-$F_c$ omit map calculated before modeling the 5′-exon-like oligonucleotide substrate 5′-AUUUAU-3′ and the compound shows that the oligonucleotide is absent. The inhibitor is instead present in the active site, as demonstrated by our crystal structure of OiD1-5 co-crystallized with the 5′-exon-like oligonucleotide 5′-AUUUAU-3′ in complex with the di-brominated intronistat B derivative (ARN25850) after 1 h soaking time (RMSD = 0.5 with respect to the intronistat B-bound structure). The anomalous difference Fourier electron density map obtained collecting the data at λ = 0.92 Å shows two pronounced peaks (highest contour value = 7.3 σ and 8 σ, respectively) in the active site at the expected *ortho* positions on the pyrogallol moiety (Fig. 4C), unequivocally confirming the binding of inhibitor in the intron active site at this step of the catalytic cycle. These data demonstrate that intronistat B competes with the 5′-exon for binding the intron active site after the first splicing step.

Interestingly, after 2.5 h soaking the crystal structure of OiD1-5 co-crystallized with the 5′-exon-like oligonucleotide and with intronistat B also reveals an $F_o$-$F_c$ omit map compatible with the binding of both oligonucleotide and inhibitor in the active site (Fig. 4E, highest contour value = 4.0 σ, volume of the peak at 3.0 σ = 301 Å³). Unexpectedly, though, this volume is oriented differently from the volume occupied by intronistat B after 1 h soaking and in the exon-free state (RSCC = 0.90, Fig. 2B, C). The structure shows that the group II intron active site still recognizes the compound's pyrogallol motif through M1 (1.9 Å from intronistat B O22 atom and 2.1 Å from intronistat B O24 atom), M2 (1.9 Å from intronistat B O24 atom and 2.2 Å from intronistat B O26 atom), and its conserved catalytic residues, i.e. the catalytic triad and the 2-nucleotide bulge (Fig. 4D). Interestingly, in this structure, the nucleotides around the splice junction also contribute to the recognition of the compound, i.e. U0 [3.3 Å between intronistat B O22 atom and U0 O3′ atom and 3.1 Å between intronistat B O24 atom and U0 O3′ atom] and U2 (weak 3.9 Å contact between O18 of intronistat B and O4′ of U2). However, in this structure intronistat B has lost its interaction with A181 in the EBS1 site (15 Å between O18 in intronistat B and N1 in A181), which is now based-paired to its cognate 5′-exon nucleotide U0 (Fig. 4 and Supplementary Fig. S6B).

Two additional independent ~500 ns-long MD simulations of intronistat B bound to OiD1-5 and 5′-exon show that the compound is steadily anchored at the active site as shown by X-ray crystallography ($RMSD_{RNA}$ = 3.8 ± 0.2 Å; $RMSD_{(intronistat B)}$ = 1.8 ± 0.2 Å, Supplementary Fig. S8). Similar to the free intron structure, also in the presence of the 5′-exon the two catalytic M1-M2 ions coordinate the compound and are stabilized at a distance $d_{M1-M2}$ = 4.01 ± 0.46 Å (Fig. 4H). Additionally, during our simulations, the contact between U2 and intronistat B is reinforced by the formation of one additional π-π stacking interaction between the U2 nucleobase and the benzofuran core of the ligand ($d\pi_{U2}$-$\pi_{intronistat B}$ = 3.91 ± 1.22 Å, Fig. 4G), which further stabilizes the binding of the compound in its crystallographic pose.

Altogether, these new structures and MD simulations of the group II intron in the presence of intronistat B before and after the first step of splicing suggest that the compound competes with the scissile phosphate to inhibit this catalytic step, curiously adopting different binding conformations. We therefore decided to probe this distinctive conformation, coupled to the inhibition constants of intronistat B and previously-reported analogs, by computing the relative binding free energy (ΔΔG) of these compounds at the intron active site (details in method section and in the legend of Supplementary Fig. S11)[19,20]. To this aim, we used both intron-ligand complexes captured by X-ray crystallography in the absence (Fig. 2A) and the presence of the 5′exon (Fig. 4A). Considering the structural homology between our *O. iheyensis* and the ai5γ introns, we then correlated these relative binding free energy values to the respective experimental inhibition constants of the compounds. When considering the exon-free binding mode (Fig. 2A), our simulations' triplicates estimated a difference in binding free energy between compound 8 and intronistat B $ΔΔG_{(compound 8 – intronistat B)}$ = −4.0 ± 1.5 kcal·mol⁻¹ (Supplementary Figs. S17, S18). Nevertheless, the standard deviation of 1.5 kcal·mol⁻¹ associated to the ΔΔG is larger than expected, likely because of poor convergence linked to enhanced flexibility of the compound when bound to the exon-free intron (Supplementary Fig. S17). On the contrary, when starting from the exon-bound binding mode, our alchemical free energy calculations estimated a well-converged $ΔΔG_{(compound 8 – intronistat B)}$ = −1.12 ± 0.70 kcal·mol⁻¹ (Fig. 5 and Supplementary Figs. S12, S13). Importantly, this ΔΔG value agrees well with the experimentally derived ΔΔG ($ΔΔG_{(compound 8 – intronistat B)}$ = −0.54 ± 0.04 kcal·mol⁻¹). This indicates that the exon-bound structure constitutes the proper model for intronistat B binding, corroborating an analogous binding mode of our intronistat B derivatives to homologous group II introns.

To further validate this distinctive binding pose, we then compared three additional pairs of previously-reported intronistat B

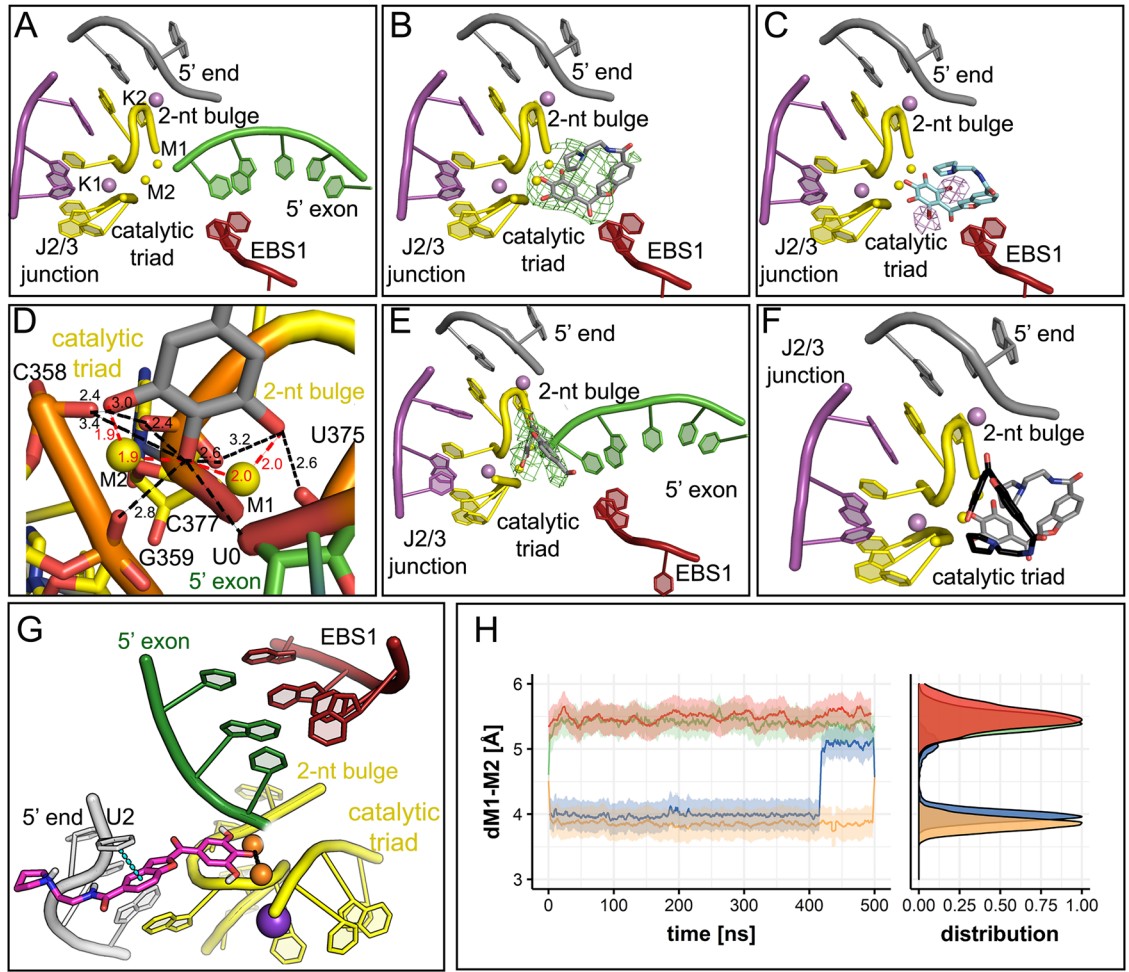

**Fig. 4 | Binding of intronistat B to the intron active site after the first step of splicing. A** Crystal structure of OiD1-5 with Mg²⁺ (yellow spheres), K⁺ (purple spheres), and the 5′-exon-like oligonucleotide 5′-AUUUAU-3′ (green sticks). Active site elements, i.e., the J2/3 junction (magenta), catalytic triad (yellow), 2-nucleotide bulge (yellow), and EB (firebrick) are shown as cartoons. **B** Crystal structure of OiD1-5 with Mg²⁺, K⁺, 5′-exon-like oligonucleotide 5′-AUUUAU-3′ (not resolved), and intronistat B (gray sticks) after 1 h soaking. $F_o - F_c$ electron density omit map calculated before modeling intronistat B, contoured at 3σ, is reported as green mesh. Representation of intronistat B color-coded by crystallographic B-factor is in Supplementary Fig. S27. **C** Crystal structure of OiD1-5 with Mg²⁺, K⁺, 5′-exon-like oligonucleotide 5′-AUUUAU-3′ (not resolved), and di-brominated intronistat B derivative ARN25850 (cyan sticks) after 1 h soaking. Violet mesh represents anomalous difference Fourier map contoured at 5σ, revealing bromide atom positions. **D** Crystal structure of OiD1-5 with Mg²⁺, K⁺, 5′-exon-like oligonucleotide 5′-AUUUAU-3′, and intronistat B after 2.5 h soaking. Interactions of intronistat B

with active site elements shown as black dotted lines for H-bonds and red for coordination bonds. **E** Crystal structure of OiD1-5 with Mg²⁺, K⁺, 5′-exon-like oligonucleotide 5′-AUUUAU-3′, and intronistat B after 2 h 30′ soaking. $F_o - F_c$ electron density omit map contoured at 3σ shown. **F** Superposition of intronistat B binding mode after 1 h soaking (gray sticks) and after 2 h 30′ soaking (black sticks). **G** MD simulations show that, in the presence of the 5′exon, intronistat B coordinates both catalytic M1-M2 ions and engages one additional π-π interaction with U2 nucleotide of the intron. Representation follows that of panel **A**. **H** Internuclear distance between catalytic M1-M2 ions reported as function of simulation time. The standard deviation is shown as a shaded area. In both free (green) and 5′exon-bound (red) intron apo-system, metals are stabilized at a distance not compatible with catalysis. In presence of intronistat B, in both free (blue) and 5′-exon-bound (yellow) intron systems, two metals are coordinated by compound (panel **A** and **C**, respectively), mimicking pseudo-Michaelis Menten complex even in absence of native substrates, rigidifying active site in non-productive conformation.

analogs [compound 8 *vs* compound 12 (APY081), $K_{i(12)} = 5.3 \pm 0.2$ μM, $\Delta\Delta G_{(8-12)} = 1.06 \pm 0.05$ kcal·mol⁻¹; compound 12 (APY081) *vs* compound 17 (APY097), $K_{i(17)} = 3.9 \pm 0.5$ μM, $\Delta\Delta G_{(12-17)} = -0.18 \pm 0.03$ kcal·mol⁻¹; compound 17 (APY097) *vs* intronistat A (APY101), $K_{i(intronistat\ A)} = 2.1 \pm 0.2$ μM, $\Delta\Delta G_{(17-intronistat\ A)} = -0.37 \pm 0.05$ kcal·mol⁻¹, Fig. 5 and Supplementary Figs. S14, S15[5]]. Remarkably, also these ΔΔG estimates overlap well with the ΔΔG derived from the experimental $K_i$ ($\Delta\Delta G_{(8-12)} = 1.44 \pm 0.33$ kcal·mol⁻¹; $\Delta\Delta G_{(12-17)} = -0.28 \pm 0.10$ kcal·mol⁻¹; $\Delta\Delta G_{(17-intronistat\ A)} = -0.39 \pm 0.10$ kcal·mol⁻¹).

The close correspondence between the experimental and computationally estimated ΔΔG values suggests that the intronistat B binding mode captured by our crystal structures accounts for all major interactions responsible for the experimentally-measured $K_i$ of the ligands.

## The mechanism of inhibition of the second step of splicing

To gain further insights into the preferential inhibition of the second step of splicing by group II intron inhibitors, we solved the crystal structure of OiD1-5 at 3.6 Å resolution, in the presence of intronistat B and of Mg²⁺ and Na⁺ ions. This condition forces the intron active site into a transiently inactive, so-called "toggled" conformation, which is achieved *via* rotation of residues 287–289 of J2/3 junction around their backbone, and which is mechanistically important because it favors the transition from the first to the second step of splicing[9,13].

In the presence of intronistat B, the intron retains the same overall structure as in the apo form (RMSD = 1.2 Å with respect to PDB id: 4FAX, Fig. 6A, B). In this structure, the $F_o$-$F_c$ electron density map, generated before modeling the compound, displays a peak in the active site, which is absent in the apo form, and which is compatible

**Fig. 5 | Splicing modulators. A** Chemical structures of the splicing modulators described in the text. The chemical groups of intronistat B are highlighted in green (pyrogallol moiety), blue (benzofuran core), and yellow (N-tail). The atom number and the corresponding $pK_a$ values are also reported. We note that the bioactive form of intronistat B is anionic on the OH group with $pK_a = 5.7$. **B** Relative binding free energy of splicing modulators. The ΔΔG values computed from experimental $K_i$ (ΔΔG$_{EXP}$) as well as that estimated computationally (ΔΔG$_{TI}$) are reported for each alchemical transformation.

with the size of the pyrogallol moiety of intronistat B (highest contour value = 4.3 σ, volume of the peak at 3.0 σ = 46 Å$^3$, RSCC = 0.92, Fig. 6B). This density suggests that intronistat B is bound to the active site also in the presence of sodium, albeit in a more flexible conformation than for the structures with potassium (see Figs. 2 and 4). Most importantly, a simulated-annealing omit map obtained by omitting residues 287–289 of J2/3 junction and M1/M2, besides intronistat B, unequivocally reveals that the intron adopts a triple helix and not the expected toggled conformation (Fig. 6B). In line with this observation, the catalytic ions M1 and M2, which are normally released from the active site in the toggled state, are instead still present in our structure.

To better validate the modeling of the pyrogallol motif in the electron density, we additionally solved the crystal structure of OiD1-5 in the presence of Mg$^{2+}$ and Na$^+$ ions in complex with the dibrominated intronistat B derivative (ARN25850) after 1 h soaking at a resolution of 4.8 Å (RMSD = 0.8 with respect to the intronistat B-bound structure). The presence of two anomalous difference Fourier electron density map peaks (highest contour value = 5.3 σ and 7.6 σ, respectively) in the active site confirms the binding mode of the pyrogallol moiety (Fig. 6C).

Furthermore, we have also obtained two structures of OiD1-5 in presence of Mg$^{2+}$, Li$^+$ and intronistat B or the di-brominated intronistat B derivative (ARN25850), respectively. As previously reported and similarly to Na$^+$, Li$^+$ also favors the toggled conformation in the absence of inhibitors[9] (Fig. 6D). In our structures with Li$^+$ and the inhibitors, instead, the intron active site adopts the triple helix conformation (Fig. 6E). This effect is due to the presence of the compound in the active site, as demonstrated by the F$_o$-F$_c$ omit map of intronistat B (Fig. 6E) and by the anomalous signal of the bromine atoms of the di-brominated derivative ARN25820 (two peaks with highest contour value = 5.9 σ and 8.5 σ, respectively, Fig. 6F). Overall, the structural data obtained in the presence of both Na$^+$ and Li$^+$ demonstrate that intronistat B prevents the stabilization of the toggled conformation.

While the flexibility of intronistat B prevents us from simulating the dynamics of the structures in presence of sodium and lithium, our equilibrium MD simulations of the OiD1-5 and Oi5eD1-5 structures with and without intronistat B in the presence of potassium provide a precise mechanistic explanation for how the compound inhibits active site toggling. Specifically, multiple ~500 ns-long MD replicas show that the presence of intronistat B prevents the formation of the catalytically-essential interaction between the active site potassium ion K1 and the N7 atom of the conserved G288 nucleotide [Fig. 6H[13],]. In the absence of intronistat B the K1-N7 interaction is properly established ($d_{K1-N7} = 3.12 \pm 0.43$ Å and $d_{K1-N7} = 3.07 \pm 0.52$ Å for OiD1-5 and Oi5eD1-5, respectively, Fig. 6G). Instead, in the presence of intronistat B the active site dynamics are altered, preventing the engagement of K1 with G288 N7 ($d_{K1-N7} = 4.72 \pm 0.53$ Å and $d_{K1-N7} = 4.68 \pm 0.61$ Å, for OiD1-5 and Oi5eD1-5, respectively, Fig. 6G, H and Supplementary Figs. S8, S9).

In summary, by impeding the establishment of the K1-N7 interaction and by preventing active site toggling, as shown by our structures in presence of sodium and lithium and MD simulations, intronistat B de-activates the intron and impairs its transition from the first to the second step.

## Discussion

Splicing is a vital and ubiquitous reaction that ensures the correct maturation of transcribed genes in all forms of life. Its modulation by small molecule compounds has recently emerged as a very promising therapeutic strategy to treat pathogenic infections, as well as human genetic diseases and cancer, but the principles by which splicing modulation is achieved are still largely unclear at the molecular level[5,6,21]. In this context, our 6 new crystal structures, corroborated by enzymatic data, atomistic simulations, and relative binding free energy calculations, now show how the evolutionarily-conserved active site of a splicing ribozyme interacts specifically and selectively with small molecule modulators (Fig. 7). Nearly 40 years after establishing that the active site of the ribosome recognizes specific small molecule inhibitors[22–24], our results exemplify the power of using RNA-binding molecules to mechanistically probe and functionally modulate another fundamental biological reaction catalyzed by RNA, splicing.

The highly-structured splicing site is perfectly suited to tightly capture and specifically recognize small organic compounds through a complex interaction network established by conserved nucleotides and the catalytic metal ion cluster. Unexpectedly, at the splicing site, the compounds display two distinct mechanisms of action. First, they compete with the substrates of the splicing reaction, i.e., the splice junctions. Second, they stabilize the intron active site in its so-called triple helix configuration, mimicking the formation of an unproductive pseudo Michaelis-Menten complex in the absence of the native

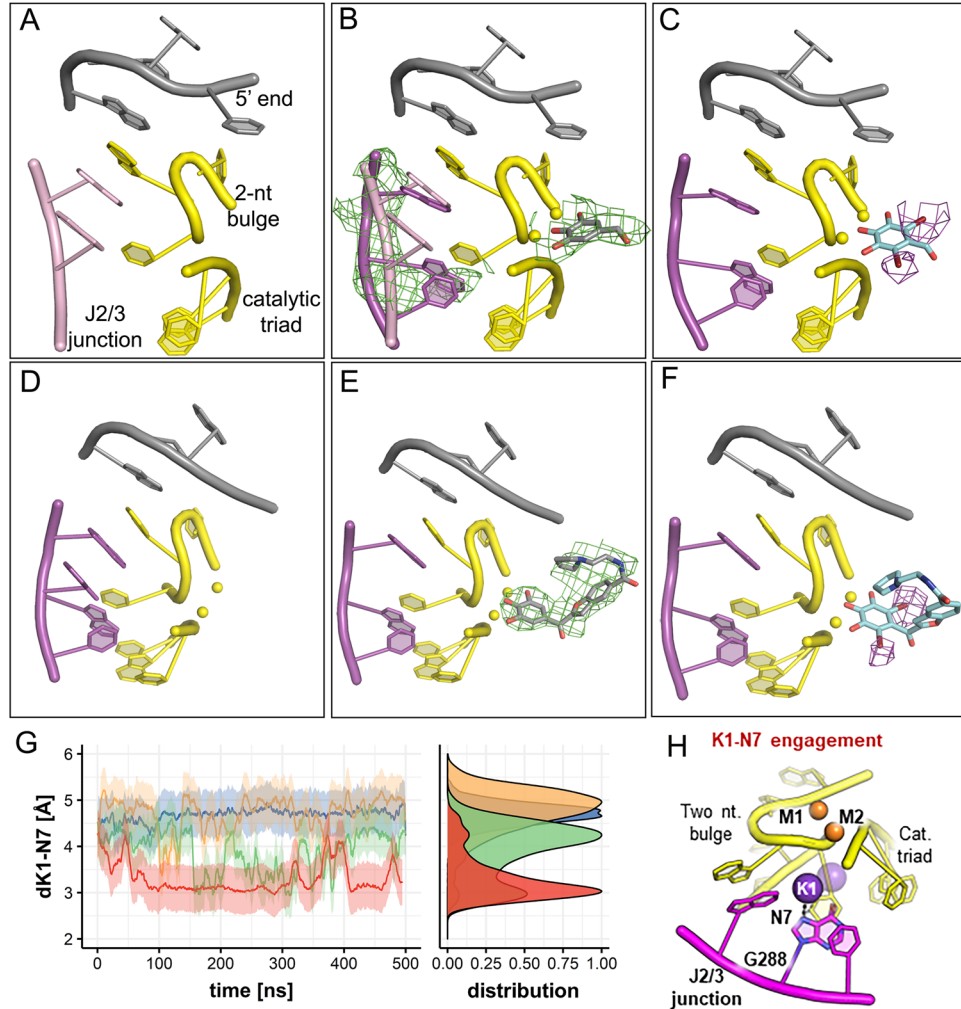

**Fig. 6 | Intronistat B prevents the toggling of the active site. A** Crystal structure of OiD1-5 with Mg²⁺ and Na⁺ (PDB id: 4FAX). **B** Crystal structure of OiD1-5 with Mg²⁺ (yellow spheres), Na⁺ and intronistat B (gray sticks, only the pyrogallol moiety is visible in the electron density). The $F_o − F_c$ electron density omit map, generated before modeling intronistat B, and the $F_o − F_c$ simulated-annealing electron density omit map, calculated by omitting the J2/3 junction residues and both M1 and M2, are shown as green mesh and contoured at 3σ. This omit map shows the J2/3 junction adopts triple-helix conformation, not toggled conformation (modeled here as pink sticks from PDB id: 4FAX). **C** Crystal structure of OiD1-5 with Mg²⁺, Na⁺, and di-brominated intronistat B derivative ARN25850 (cyan sticks). Violet mesh represents anomalous difference Fourier map contoured at 4σ, revealing bromide atom positions. **D** Crystal structure of OiD1-5 with Mg²⁺ and Li⁺. **E** Crystal structure of OiD1-5 with Mg²⁺, Li⁺ and intronistat B (gray sticks). The $F_o − F_c$ electron density omit map generated before modeling intronistat B is shown as green mesh contoured at 3σ. J2/3 junction adopts only triple-helix conformation, not toggled

conformation. **F** Crystal structure of OiD1-5 with Mg²⁺, Li⁺ and the di-brominated intronistat B derivative ARN25850 (cyan sticks). Violet mesh represents anomalous difference Fourier map contoured at 4σ, revealing bromide atom positions. **G** Distance between K1 and N7 of G288 plotted as function of simulation time. In absence of intronistat B, K1-N7 interaction engaged in both free (green) and 5'exon-bound (red) intron systems. The standard deviation is shown as a shaded area. When intronistat B coordinates catalytic M1-M2 ions, it alters functional dynamics of active site, preventing formation of K1-N7 interaction in both free and 5'-exon-bound system, impairing splicing initiation and progression. **H** Catalytic K1 ion (purple sphere) engages interaction with N7 atom of G288 (purple sticks) at J2/3 junction (purple cartoon). If ion not in place (purple semitransparent sphere), K1-N7 interaction not established, ultimately inhibiting catalytic activation of intron for first splicing step, and its structural rearrangement necessary for splicing progression. Representation follows that of panel A.

substrates. This binding mode prevents the formation of the catalytically essential K1-N7 interaction at the active site, ultimately impairing the intron from toggling into an open and transiently-inactive state after completion of the first step of splicing. Through mutagenesis, metal ion replacement approaches, and atomistic simulations, we had previously demonstrated that such transiently-inactive toggled configuration is essential for preparing the intron active site for the second step of splicing[9,13]. In line with such mechanistic data, here we observe that by impeding active site toggling, intronistat B selectively induces a severe defect of the second step.

As revealed explicitly for intronistat B, the compound's interactions with the intron are primarily sequence-unspecific, in line with the broad spectrum of the compound, which inhibits both bacterial and

organellar introns belonging to different classes (groups IIB and IIC) and following different splicing pathways (hydrolysis *vs* transesterification). These interactions are driven by the highly-specific contacts established by the pyrogallol moiety of intronistat B with the intron's metal core, with important implications for the use of these phenolic derivatives in targeting RNA-bound metal clusters. Indeed, our results allow clear rationalization of previously-reported intronistats' structure-activity relationships (SAR)[5]. For instance, compound 2 (APY007), which differs from compound 1 by lacking one hydroxyl group in meta on the pyrogallol, has an IC₅₀ of >100 μM. More generally, all reported compounds in which the pyrogallol motif is absent have IC₅₀ values > 100 μM, while all compounds in which the pyrogallol is present have IC₅₀ values in the range of 1–90 μM[5]. Furthermore,

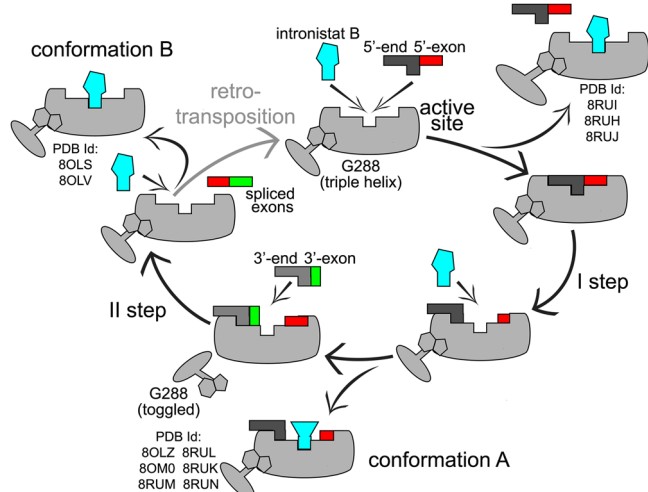

**Fig. 7 | Schematic mechanism of group II intron splicing inhibition by small molecule compounds.** In the pre-catalytic state, intronistat B competes with the 5′-splice junction at the intron active site to inhibit the first step of splicing. Intronistat B also binds the intron active site in the presence of the 5′-exon after the first step of splicing, by establishing only sequence-unspecific interactions and contacts with active site metal ions (conformation A, Fig. 4). In this state, intronistat B prevents the intron from switching from its triple-helix to its toggled conformation, thus inhibiting the second step of splicing more potently than the first. Finally, intronistat B also binds the exon-free intron by establishing contacts with the catalytic metal ions and both sequence-specific and sequence-unspecific interactions (conformation B, Fig. 2). Relevant structures (PDB ids) obtained in this work are indicated for each state.

comparison of the pyrogallol-containing compounds allows us to rationalize also the importance of the intronistat B benzofuran core and N-tail. From our structures, it appears that the N-tail, which is dynamic and thus less well-resolved in the electron density, may be important to fit into the negatively-charged environment of the RNA-based splice core. Interestingly, compound 11 (APY093)[5], which possesses a rigid, multi-aromatic core and a sulfonamide tail loses affinity to the splice core by one order of magnitude, with respect to intronistat B. Instead, the benzofuran core plastically adapts to conformational changes that occur throughout catalysis, possibly driven by weak π-π stacking interactions with dynamic nucleobases close to the splice junction (i.e. U2, Fig. 2). Importantly, the benzofuran core also establishes sequence-specific interactions with A181 in EBS1. This residue is not conserved in group II introns and it can be replaced by purines or by pyrimidines, i.e. a G in *Escherichia coli* EcI5 group IIB intron[25] and in *Lactococcus lactis* LlI1 group IIA intron[26], a C in *Thermosynechococcus elongatus* Tel4h group IIB intron[27] and *Bacillus halodurans* BhI1 group IIC intron[28], or a U in *Pylaiella littoralis* PliLSUI2 group IIB intron[29]. This observation suggests that a small molecule like intronistat B can be designed and modified to establish stronger, sequence-specific contacts with the EBS1 site to generate more selective, species-specific compounds. In this context, our computational protocol to estimate the relative binding free energy of intronistat B analogs can now be used for the rational design of such sequence-specific modulators, building upon the intronistat B-intron interaction network captured crystallographically in our work.

Quite remarkably, intronistat B not only binds tightly to the active site and displays a selective dual mechanism of inhibition, but it also dynamically adapts to conformational changes in the intron active site, by changing its binding mode while the intron progresses through its different catalytic states. At a time when the determinants that govern RNA-small molecule interactions are still largely uncharacterized, our observations of how intronistat B binds to the splice site constitutes an important example of how RNA structure

functional dynamics can be accurately modulated with small molecules.

Our data thus represent a solid experimental basis for directing the rational design of splicing modulators through structure-based strategies, in the same way that is traditionally applied to protein design and drug discovery. Mechanistic studies on the mode of action of ribosomal RNA inhibitors provided important insights for developing new species-specific antimicrobial agents overcoming the insurgence of resistance, besides crucially improving our understanding of the process of translation[24]. Similarly, our study could help improve the potency of intronistat B, which is still insufficient to make this tool compound a suitable drug lead for the treatment of fungal infections (low micromolar $K_i$ and $IC_{50}$ values). Potentially, group II intron inhibitors could also have relevance in the context of bacterial infections. In bacteria, group II introns are often located in plasmids associated to genes that confer antibiotic resistance[30,31]. Although a clear connection between group II intron activity and bacterial infectivity is still lacking, it cannot be excluded that inhibiting bacterial splicing could prevent antibiotic resistance. Certainly, though, bacterial group II introns act as retroelements[32] and thus have established biotechnological applications in genome editing, as so-called "targetrons"[33]. As such, bacterial group II introns can be used to site-specifically deliver cargo genes into genomic sites, including genes for fluorescent proteins, phage resistance, and antigens. However, targetrons have limited applicability because of their off-target effects and poor specificity. By establishing that the exon-free state of group II introns (i.e. the state that catalyzes retro-homing) binds small molecules tightly and in a sequence-specific manner, our study now opens up the possibility of developing improved targetrons. For instance, it could be envisaged to design targetrons that are constitutively inhibited by photoactivatable intronistat B derivatives, deliver these ribozymes systemically, and only photoactivate them in the desired target tissues, thus reducing their toxicity[34,35].

Most importantly, by proving that the conserved RNA-core of self-splicing ribozymes is capable of recognizing small molecules specifically and selectively, our work sets a solid rationale for the design of analogous compounds to also inhibit the eukaryotic spliceosome. The spliceosome active site derives evolutionarily from that of group II introns and preserves its key structural, chemical and functional properties[11]. Intronistat B is not active against the spliceosome[5]. However, by superimposing the intron and spliceosome active site structures, we note that compounds analogous to intronistat B could fit within the catalytic core of spliceosomal complexes formed at different stages of catalysis, including the B* complex (i.e. before the first step of splicing, PDB id: 5Z56, 5Z57, 5Z58) and the $C_i$ complex (i.e. after the first step of splicing, PDB id: 7B9V). Here, the compounds could anchor at the M1-M2-K1 site with their pyrogallol or an analogous moiety, as our work now proves possible. In this binding pose, the compounds would be able to explore multiple conformations, interacting with catalytic nucleotides, metal ions or protein splicing factors (Supplementary Fig. S16). Importantly, by binding at the active site, the compounds would also necessarily reach close to nucleotides around the splice junctions. Establishing direct interactions with pre-mRNA nucleotides close to the splice junctions would crucially offer unprecedented opportunities for sequence-specific and gene-selective modulation of splicing. Such selectivity cannot be achieved by targeting splicing assembly factors, a splicing modulation strategy that is currently widely explored, but that inevitably causes severe toxicity[21,36]. Importantly, achieving gene-selective splicing modulation with small molecule inhibitors could also offer advantages over the established use of splice-switching oligonucleotides (SSO), which are expensive, display poor cellular uptake and tissue/cell-specificity, and are prone to off-target effects[37]. More broadly, considering that the group II intron and spliceosome active site architecture and catalytic mechanism are also very similar to those of nucleic acid-processing

protein enzymes, like DNA/RNA polymerases and endo/exonucleases, it should not be excluded that the design of specific inhibitors against one of these convergently-evolved enzymes could then inform the rational design of analogs that inhibit other target classes[38–40].

In summary, our data demonstrate how the conserved RNA-based splice site recognizes small molecule modulators. These RNA-targeting compounds thus emerge as very informative tools to mechanistically probe a vital biological reaction, such as splicing, and to foster the design of new biotechnological, genetic engineering and pharmacological applications.

## Methods

### Cloning and mutagenesis
In this work, we have used two different introns (secondary structures reported in Supplementary Fig. S1). The *S. cerevisiae* ai5γ group IIB intron and the *O. iheyensis* I1 group IIC intron. For the *S. cerevisiae* ai5γ group IIB intron, we have used one single construct, formed by domains 1-3 and 5 (D135), which had been previously used for determining the intronistat B $IC_{50}$ using a FRET-based spliced-exon reopening assay[5]. Instead, for the *O. iheyensis* I1 group IIC intron, we have used four constructs, as follows. We have used the wild-type *O. iheyensis* group II intron sequence (479 nt) flanked by its 5' exon (223 nt) and by its 3' exon (101 nt), corresponding to the previously-described pOiA and pOi5 constructs[9,13,41] for the analysis of splicing kinetics. We have used the previously-described OiD1-5 construct[9,13,41,42] to crystallize the intron in the substrate-free state. We have used the previously-described Oi5eD1-5 construct, corresponding to OiD1-5 following the 5'-exon 5'-UUAU sequence[9], to crystallize the intron in the pre-catalytic stage. Finally, we have generated a new crystallization construct corresponding to OiD1-5 without the three 5'-terminal Gs to crystallize the intron in the 'mimic' of the post catalytic stage. We deleted the three 5'-terminal Gs from OiD1-5 by Gibson assembly using the Gibson Assembly Kit (NEB). We have used the restriction enzyme ClaI (NEB) for linearization of pOiA templates, and BamHI (NEB) for linearization of D135 and of all templates of the crystallization constructs. We have confirmed the identity of all constructs by DNA sequencing (Eurofins).

### In vitro transcription and purification
All constructs were linearized by digestion with the appropriate endonucleases at 37 °C overnight and transcribed in vitro using T7 polymerase[9]. The constructs used for the crystallography experiments were then purified under native conditions[43], buffered again and concentrated to 80 μM in 10 mM MgCl$_2$ and 5 mM sodium cacodylate pH 6.5. The construct used for splicing assays was radiolabeled during transcription, purified to a denatured state[9], and subsequently refolded[41]. The construct used for the FRET assay was purified to a denatured state and refolded just before the assay[5].

### Splicing assays
Purified radiolabeled intron precursor was refolded by denaturation at 95 °C for 1 min in the presence of 40 mM Na-MOPS pH 7.5, and cooled at room temperature for 2 min. Then KCl to a final concentration of 150 mM and MgCl$_2$ to a final concentration of 5 mM were added[9,13], and the refolded intron was incubated with various concentrations of intronistat B [N-(2-(pyrrolidin-1-yl)ethyl)-2-(3,4,5-trihydroxybenzoyl) benzofuran-5-carboxamide; Sigma Aldrich catalog number SML2630] or ARN25850 (self-synthesized, see below) at 37 °C. Aliquots at different time points were quenched with urea and analyzed on a 5% denaturing polyacrylamide gel[44]. The kinetic rate constants ($k_{obs}$) in presence of each compound concentration were calculated using the Prism 8 package (GraphPad Software) and plotted against the inhibitor concentration according to the following equation to determine $K_i$ values: $k_{obs} = k_{max}/(1 + [I]/K_i)$, where $k_{obs}$ and $k_{max}$ are the rate constants measured in the presence and in the absence of the inhibitor,

respectively, [I] is the concentration of the inhibitor and $K_i$ is the inhibition constant[5]. Experiments were performed in triplicate. Data represent average ± s.e.m. The $IC_{50}$ value was calculated by measuring the fraction of precursor (5e-I-3e) at the reaction time of 15 min ($F_{5e-I-3e, 15min}$), calculating the percentage of reacted precursor at each concentration of compound according to the following equation: $\%_{5e-I-3e} = 100*(F_{5e-I-3e, 15min, [I]max} - F_{5e-I-3e, 15min, [I]})/(F_{5e-I-3e, 15min, [I]max} - F_{5e-I-3e, 15min, DMSO})$, and fitting the percentage of reacted precursor as function of the compound concentration according to the following function: $\%_{5e-I-3e} = 100/(1 + [I]/IC_{50})$. Data are reported as average ± s.e.m.

### Fluorescence resonance energy transfer (FRET)-based spliced-exon reopening (SER) assay
The FRET-based SER assay was performed as previously reported[5]. Purified *S. cerevisiae* D135 group IIB intron was refolded by denaturation at 95 °C for 1 min in the presence of 50 mM Na-MOPS pH 7.0, and cooled to room temperature for 2 min. Then KCl to a final concentration of 500 mM and MgCl$_2$ to a final concentration of 100 mM were sequentially added. 96-well PCR plates (Azenta Life Sciences) were filled with 10 μl of solution containing 20 nM *S. cerevisiae* D135 group IIB intron, 20 nM of a double-labeled RNA substrate previously reported (Integrated DNA Technologies)[5] and small molecules in 50 mM MOPS, pH 7.0, 100 mM MgCl$_2$ and 500 mM KCl. Intronistat B and ARN25850 were tested at 11 different concentrations ranging from 250 nM to 1 mM. Each measurement was performed in triplicate. Plates were incubated at 37 °C for 60 min in an RT-PCR machine (Biorad). Each experiment was performed in triplicate. The $IC_{50}$ value was calculated by measuring the initial slope of the response curves and by calculating the percentage of intron activity in presence of each compound concentration with respect to the initial slope of the reaction in presence of only DMSO. Data were fit using the Prism 8 package (GraphPad Software) and plotted against the inhibitor concentration according to the following equation: $\%_{activity} = 100/(1 + [I]/IC_{50})$. Data are reported as average ± s.e.m.

### Isothermal titration calorimetry (ITC)
All experiments were carried out using a MicroCal iTC200 instrument (Malvern) by using the Oi1-5 construct. Experiments were performed at 25 °C in 10 mM MgCl$_2$, 40 mM MOPS-NaOH pH 7.5, 150 mM KCl and 1% DMSO. The intron solution at the concentration of 30 μM was loaded in the calorimetric cell. Intronistat B at the concentration of 600 μM was titrated in the intron sample by performing 16 injections of 2.5 μL aliquots. A blank control experiment was performed by injecting intronistat B into the calorimetric cell containing only the reaction buffer. The dissociation constants ($K_D$), enthalpy of binding ($\Delta H$), and binding stoichiometry (N) were obtained after fitting the integrated data, normalized by subtracting the blank control, to a single-site binding model. The compound concentration was adjusted to maintain a 1:1 stoichiometry. Data were processed using MicroCal PEAQ-ITC Analysis Software. All experiments were performed in triplicate.

### Bio-layer interferometry (BLI)
Measurements were performed on the OctetRED96e (Sartorius) using High Precision Streptavidin (SAX) Biosensors (Sartorius) and were recorded with the manufacturer software (Data acquisition v11.1). Experiments were performed by using the Oi1-5 construct biotinylated at the 3'-end as following. First, 20 μM Oi1-5 construct was oxidized at the 3'-terminal ribose with 100 mM sodium periodate in 100 mM Na-acetate pH 5 at 25 °C, 1000 rpm for 90 min in the dark. Then the oxidized construct was reacted with 100 mM EZ-link hydrazide PEG-4 Biotin at 25 °C, 1000 rpm for 4 h. The biotinylated RNA was then purified and stored in denatured state in order to be refolded before the assay by denaturation at 95 °C for 1 min in the presence of 40 mM Na-MOPS pH 7.5, and cooled at room temperature for 2 min. Then KCl to a

final concentration of 150 mM and $MgCl_2$ to a final concentration of 5 mM were added and the folded intron used for the assay. BLI analysis were performed in 0.2 ml solution per well in black 96-well plates (Nunc F96 MicroWell, ThermoScientific) at 25 °C and at 1000 rpm agitation. The streptavidin biosensors were hydrated with analysis buffer (10 mM $MgCl_2$, 40 mM MOPS-NaOH pH 7.5, 150 mM KCl, 0.02 % TWEEN-20 and 1% DMSO) for 10 min, followed by equilibration in the same buffer for 120 s and loaded with a solution 100 nM biotinylated Oi1-5 for 10 min to reach a spectral shift of 1 nm. After the intron loading, the functionalized biosensors were saturated by dipping them in a 100 mM biocytin solution in order to avoid any unspecific binding of the compound to streptavidin. For kinetics measurements, intronistat B was serial diluted in analysis buffer at concentrations between 12.5 and 100 µM. Association phases were monitored by dipping functionalized biosensors in analyte solutions for 300 s, and the dissociation phased monitored in analysis buffer for 300 s. Kinetics data were processed with the manufacturer software (Data analysis HT v11.1) by subtracting the signal from a zero-concentration sample and baseline-aligned using an additional equilibration step of 120 s in analysis buffer performed prior the association step. Specific kinetics signals were then fitted using a global fit method and a 2:1 heterogeneous model. Affinity constant was obtained by the ratio of kinetics rate constants. Reported values were obtained by averaging values obtained with duplicated and independent assays and errors as the standard deviation.

## Crystallization

The natively purified intron was mixed with a 0.5 mM spermine solution and with the crystallization buffer in a 1:1:1 volume ratio[9]. Crystals at the 'mimic' of the post-catalytic stage were crystalized in presence of a substrate oligonucleotide 5′-AUUUAU-3′ 100 µM. Crystals were grown at 30 °C by the hanging drop vapor diffusion method using 2 µL sample drops and 300 µL crystallization solution in a sealed chamber (EasyXtal 15-Well Tool, Qiagen). Crystals were soaked for 1 h (for the Na structure or with the dibromo Intronistat B derivative) or 2 h 30′ (for the structures in the pre-catalytic, post-catalytic, 'mimic' of the post-catalytic and the exon-free stages) in a solution containing the corresponding crystallization buffers supplemented with 1 mM intronistat B or ARN25850 and cryo-protected with 25% ethylene glycol before flash freezing in liquid nitrogen. Crystals were harvested after 2–3 weeks, except those used to solve the structures in the prehydrolytic states, which were frozen within 24 h. The crystallization solutions used to solve the structures presented in this work were composed of: (1) 100 mM K-Acetate, 100 mM KCl, 100 mM $CaCl_2$, 50 mM Na-HEPES pH 7.0, 3% PEG 8000 for the calcium data set representing the pre-catalytic stage; (2) 100 mM Mg-Acetate, 150 mM KCl, 10 mM LiCl, 50 mM Na-HEPES pH 7.0, 4% PEG 8000 for the potassium data set representing the post-catalytic stage; (3) 100 mM Mg-Acetate, 200 mM KCl, 50 mM Na-HEPES pH 7.0, 6% PEG 8000 in the presence of the 6-mer RNA 5′-AUUUAU-3′ for the potassium data sets representing the 'mimic' of the post-catalytic stage; (4) 100 mM Mg-Acetate, 150 mM NaCl, 50 mM Na-HEPES pH 7.0, 5%PEG 8000 for the sodium data sets; (5) 100 mM Mg-Acetate, 200 mM KCl, 50 mM LiCl, 50 mM Na-HEPES pH 7.0, 4% PEG 8000 for the potassium data set representing the substrate-free state; and (6) 100 mM Mg-Acetate, 150 mM LiCl, 50 mM Na-HEPES pH 7.0, 5%PEG 8000 for the lithium data sets.

## Structure determination

Diffraction data were collected at beamlines ID30A-1 and ID30B at ESRF (Grenoble, France)[45,46]. Data collection for the dibromo Intronistat B derivative was performed at 0.92 Å and optimized to maximize measurement of the bromine anomalous signal. All data was processed with the XDS suite[47]. The structures were solved by molecular replacement using Phaser in CCP4[48] and the RNA coordinates of PDB entry 4FAR and 4E8M (without solvent atoms) as the initial model[9,49,50].

The models were improved automatically in Phenix[51–53] and Refmac5[54] and manually in Coot[55], and finally evaluated by MolProbity[56]. The figures depicting the structures were drawn using PyMOL Molecular Graphics System (Schrödinger).

## $pK_a$ calculations

Using MacroModel (v11.2, Schrodinger), we performed an OPLS4-based[57] conformational search of the input ligand structure and included in the next calculation the five most stable conformations of both ligand's protonation states, for a total of 10 input conformations. Then, we employed the fully-analytic DFT-based protocol implemented in the Jaguar $pK_a$ computational tool with default parameters, allowing the use of zwitterionic functional groups[58].

## Equilibrium molecular dynamics simulations

We used four structural models to perform MD simulations. Specifically, we modeled the holo-systems based on the structure of the intronistat B bound to the free intron (Fig. 2, PDB id: 8OLS), and that of the ligand in complex with the 5′exon-bound intron (Fig. 4, PDB id: 8OLZ). We modeled the corresponding apo-systems by removing the ligand from the holo-models, thus obtaining the free intron and 5′-exon-bound intron structures.

We used the AMBER force-field RNA.OL3 to parametrize the RNA molecules[59,60]. We used Joung and Cheatham[61], Panteva 12-6-4 non-bonded fixed-point charge[62], and TIP3P models[63] to parametrize ions, divalent ions and water, respectively.

We used the following protocol to set up and perform the production run for all the structural models. We performed Langevin dynamics simulations[64] with an integration time step of 2 fs, using AMBER[65]. We set the pressure at 1 atm using a Berendsen barostat with a relaxation time of 2 ps, while the temperature of 300 K was controlled using a collision frequency γ = 1 per ps[66].

First, we performed energy minimization to relax the bulk water molecules, imposing 300 kcal·mol·$Å^2$ on the heavy atoms of the models, including the crystallized metals. Then, we smoothly thermalized the whole system at 300 K using one NVT simulation of ~1 ns, keeping the positional restraints used during the energy minimization. We then halved such restraints with a series of 3 ~300-ps-long NVT simulations, performed two additional NPT simulations to relax the density of the systems to ~1.01 g·$cm^{-3}$, and further halved the restraints, which we removed during a third NPT run of ~2 ns. We performed 2 independent simulation replicates for each structural model, resulting in 8 independent MD simulations and a total simulation time of 4 µs.

## Thermodynamic integration-based alchemical free energy calculations

Alchemical free energy calculations based on thermodynamic integration (TI) allow the estimation of the relative free energy of binding of two ligands, namely A and B, from molecular simulations that connect A and B bound or unbound states via a thermodynamic cycle (Supplementary Fig. S11A). This cycle is created by two thermodynamic paths that "alchemically" bridge the bound-A to bound-B states and unbound-A to unbound-B states, respectively, via a series of unphysical states. This is possible by introducing a coupling parameter λ that varies from zero (i.e., ligand A) to one (i.e., ligand B) and controls the unphysical decoupling of the contributions to the potential energy (i.e., van der Waals and Coulomb intra- and intermolecular interactions) of each ligand from the bound state and recouple them in the unbound state (water). In practice, two series of simulations are performed. One set of simulations alchemically transforms the ligand A into B when they are immersed in bulk water, thus representing the unbound state. The same is done for the bound state, thus transforming the ligand A into B when they are in complex with the receptor. Then, the ΔG associated with

these alchemical transformations is estimated ($\Delta G_A$ and $\Delta G_B$, Supplementary Fig. S11A), and, building upon the thermodynamic cycle in Supplementary Fig. S11A, it is possible to calculate $\Delta\Delta G_{AB} = \Delta G_B - \Delta G_A$. A detailed description of the computational method and its theoretical framework is out of the scope of this study and can be found elsewhere[19].

So far, alchemical free energy calculations have been reliably used to guide the structure-based design of small molecule drugs and predict the effect of functional groups on the potency of the newly designed compound before it is synthesized and tested[67,68]. However, for alchemical free energy calculations to be effectively reliable, the binding mode of the ligands must be modeled very accurately to account for all the interactions responsible for the experimentally observed binding potency. Otherwise, the free energy estimates of the bound state could be dramatically affected, resulting in an inaccurate estimate of the total $\Delta\Delta G_{AB}$[69,70]. Building upon this evidence, in this study we decided to use alchemical free energy calculations in a reverse approach than usually done. That is, instead of relying on a validated binding mode to estimate compounds' $\Delta\Delta G_{AB}$ and computationally drive the improvement of binding potency, here we relied on experimentally computed $\Delta\Delta G_{AB}$ of small molecules to compare with our estimates so as to computationally validate the ligands' binding mode.

We employed the GPU-TI code implemented in AMBER22 to perform alchemical free energy calculations[65,71]. The ligand-bound structural models were prepared as follows. Given a couple of ligands A and B, where B is bigger than A, then the binding of ligand A was modeled at the active site of the intron building upon the pose of intronistat B. This has been possible since we evaluated $\Delta\Delta G$ only of ligands having the same scaffold as intronistat B (i.e., the pyrogallol moiety and the benzofuran core). The ligand-A bound complex was then equilibrated via classical MD simulations using the same protocol mentioned in the previous paragraph, and a production run of 100 ns was performed to properly relax the system (Supplementary Fig. S11B). The last frame's coordinates obtained via this MD simulations were used to set up the topology for the TI-based alchemical transformations, which were performed disabling the SHAKE algorithm, setting the time step to 1 fs and the collision frequency to 5, as well as using the Monte Carlo barostat with a pressure relaxation time of 1.0 ps to keep the pressure at 1.01325 bar. TI simulations followed a $\lambda$-scheme, similar to previous works (Supplementary Fig. S11B)[72], starting with a preliminary thermalization (1 ns) and pressurization (1.5 ns) of the system in the NVT and NPT ensemble, respectively, setting $\lambda = 0.5$. At this point, a divergent equilibration scheme is initiated to equilibrate all twelve $\lambda$-windows ($\lambda$: 0.00922, 0.04794, 0.11505, 0.20634, 0.31608, 0.43738, 0.56262, 0.68392, 0.79366, 0.88495, 0.95206, and 0.99078 and weighs 0.02359, 0.05347, 0.08004, 0.10158, 0.11675, 0.12457) via two parallel branches of NPT simulations (2.5 ns, each $\lambda$-window). One branch leads to the progressive equilibration of the windows from $\lambda = 0.5$ to $\lambda = 0.00922$, the other from $\lambda = 0.5$ to $\lambda = 0.99078$, such that each simulation is started from an equilibrated frame taken from the previous $\lambda$-window. Finally, production simulations are performed using Hamiltonian replica exchange via $\lambda$-hopping, with an exchange rate of 10 ps, for a total simulation time of 5 ns per $\lambda$-window.

For the ligand-in-water structural models, the compounds were immersed in a box of TIP3P water whose edges were separated by 20 Å from the solute, and monovalent ions were added to neutralize the system when needed. We used the same simulation protocol described for the ligand-bound state. For both the bound and unbound ligands' states, in case of non-neutral transformations (i.e., the net charge of one compound differs from the one of the other), we applied the so-called co-ion approach[73,74]. Here, one metal ion with appropriate charge is transformed into one water molecule to balance the net charge of the system during the compounds' transformation. Smoothstep softcore potentials were used to treat ligands' non-common atoms[75].

Overall, we performed the following TI-based alchemical free energy calculations, cumulating ~1μs of simulation time: *i)* compound 8 → intronistat B as bound to the free intron (Supplementary Figs. S17, S18, 3 replicas); *ii)* compound 8 → intronistat B as bound to the intron in complex with the 5'exon (Figs. S12, S13, 3 replicas); *iii)* compound 8 → compound 12 as bound to the intron in complex with the 5'exon (Supplementary Figs. S14A, S15A, 1 simulation); *iv)* compound 12 → compound 17 as bound to the intron in complex with the 5'exon (Supplementary Figs. S14B, S15B, 1 simulation); *v)* compound 17 → intronistat A as bound to the intron in complex with the 5'exon (Supplementary Figs. S14C, S15C, 1 simulation). All compounds have been modeled with deprotonated *p*-OH group at pyrogallol moiety, as supported by QM/MM calculations (Supplementary Figs. S24, S25).

### Hybrid quantum mechanical/molecular mechanical (QM/MM) simulations

QM/MM simulations[76–78] were performed with the QUICK code[79,80] interfaced with the MD engine SANDER available from AmberTools[76,77,81]. Simulations were performed for ligands intronistat B (both deprotonated and protonated states were considered) and its brominated derivative. Models were constructed from the respective experimental structures as bound to the exon-free intron (PDB 8OLS and 8OLV, respectively). The QM region included part of the ligand (the pyrogallol and benzofuran moieties) and the magnesium ions with their full coordination sphere, accounting for about 100 atoms and a total charge of −1 (Supplementary Figs. S24, 25). QM atoms were described by the hybrid B3LYP[82,83] DFT functional and basis set 6-31G(d,p)[84,85]. The remaining atoms were treated at the MM level as described in the paragraphs above. The QM/MM simulations were performed at constant volume and temperature (300 K) using a time step of 0.5 fs to integrate the equations of motion. QM/MM simulations were started from preliminary classical MD simulations.

### Chemistry

**General considerations.** All the commercially-available reagents and solvents were used as purchased from vendors without further purification. Dry solvents were purchased from Sigma-Aldrich. Intronistat B hydrobromide (≥98%) was purchased from Sigma-Aldrich and used for the assays reported in the present work. A separate batch of intronistat B hydrobromide was synthesized as described in WO2019147894, and used as a starting material for the synthesis of ARN25850 [2-(2,6-dibromo-3,4,5-trihydroxybenzoyl)-*N*-(2-(pyrrolidin-1-yl)ethyl)benzofuran-5-carboxamide hydrobromide, Supplementary Fig. S29]. NMR data were collected on a Bruker Avance III 400 MHz ($^1$H) and 100 MHz ($^{13}$C). Spectra were acquired at 300 K, using deuterated dimethylsulfoxide (DMSO−$d_6$). For $^1$H-NMR, data are reported as follows: chemical shift, multiplicity (s= singlet, d= doublet, dd= double of doublets, t= triplet, q= quartet, m= multiplet), coupling constants (Hz) and integration. UPLC/MS analyses were run on a Waters ACQUITY UPLC/MS system consisting of an SQD (Single Quadrupole Detector) Mass Spectrometer equipped with an Electrospray Ionization interface and a Photodiode Array Detector. PDA range was 210−400 nm. The mobile phase was 10 mM NH$_4$OAc in H$_2$O at pH 5 adjusted with AcOH (A) and 10 mM NH$_4$OAc in CH$_3$CN-H$_2$O (95:5) at pH 5 (B). The analysis was performed on an ACQUITY UPLC BEH C18 (100 × 2.1 mmID, particle size 1.7 μm) with a VanGuard BEH C18 pre-column (5 × 2.1 mmID, particle size 1.7 μm) and the mobile-phase B proportion increased from 10 % to 90 % in 6 min. Electrospray ionization in positive and negative mode was applied in the mass scan range 100−650 Da.

**Synthesis of di-brominated intronistat B derivative, ARN25850.** 1,3-Dibromo-5,5-dimethylhydantoin (DBDMH) (48.5 mg, 0.17 mmol) was added portionwise to a suspension of intronistat B hydrobromide (60.0 mg, 0.12 mmol) in CHCl$_3$ dry (3.5 mL) at room temperature under argon. The resulting suspension was stirred for 72 h; the limpid

supernatant was decanted from the solid material. This latter was triturated with isopropanol (1 mL), followed by 10% isopropanol in chloroform (twice), and dichloromethane (once). The resulting solid was dried *in vacuo* (60 °C) to provide the title compound as a brownish solid (16 mg, 20%). UPLC/MS: Rt: 2.50 min. MS (ESI) m/z: 566.97 [M-H]⁻. [M-H]⁻ calculated for $C_{22}H_{20}Br_2N_2O_6 = 566.96$, (Supplementary Fig. S19). $^1H$ (400 MHz, DMSO-$d_6$): δ 9.70 (bs, 2H), 9.50 (bs, 1H), 8.86 (t, $J = 5.2$ Hz, 1H), 8.35 (d, $J = 1.8$ Hz, 1H), 8.09 (dd, $J = 8.9, 1.8$ Hz, 1H), 7.87 (d, $J = 8.8$ Hz, 1H), 7.70 (s, 1H), 3.64 (m, 4H), 3.08 (m, 2H), 2.02 (m, 2H), 1.87 (m, 2H). (One $CH_2$ overlaps with water, signal recovered by HSQC, 3.37ppm). $^{13}C$ NMR (101 MHz, DMSO-$d_6$) δ 183.0 (Cq), 166.4 (Cq), 157.0 (Cq), 151.9 (Cq), 143.6 (Cq, 2C), 137.0 (Cq), 130.5 (Cq), 129.8 (Cq), 128.4 (CH), 126.8 (Cq), 124.0 (CH), 117.7 (CH), 112.4 (CH), 98.7 (Cq, 2C), 53.6 ($CH_2$, 2C), 53.3 ($CH_2$), 36.0 ($CH_2$), 22.5 ($CH_2$, 2C). $^1H$, $^{13}C$, COSY and HSQC NMR spectra and MS and VIS-UV spectra are represented in Supplementary Figs. S19–S23.

### Reporting summary

Further information on research design is available in the Nature Portfolio Reporting Summary linked to this article.

## Data availability

Coordinates and structure factors have been deposited in the Protein Data Bank under accession codes 8OLS, 8OLV, 8OLW, 8OLY, 8OLZ, 8OM0, 8RUH, 8RUI, 8RUJ, 8RUK, 8RUL, 8RUM and 8RUN. Data supporting the findings of this study are available within the paper and its Supplementary Information files. Source data are provided with this paper.

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

## Acknowledgements

We thank the scientists of the ESRF and EMBL Grenoble for assistance and support in using beamlines ID30-A1 and ID30B under proposal numbers MX-2532 and MX-2458. We thank Dr Isabel Chillon (CNRS, IGMM Montpellier) for help with initial splicing kinetic assays, and Ombeline Pessey and Dr Ines Dieryck for excellent technical assistance. We thank Prof Anna Pyle and Dr Olga Fedorova for advice on ITC and BLI and for critical feedback of the manuscript. We also thank Prof Annalisa Pastore, Prof Kristina Djinovic-Carugo, Dr Isabel Chillon, Dr Michela Nigro and Shekhar Jadhav for critical reading of the manuscript. We thank all members of the Marcia and De Vivo labs for helpful discussion. Work in the Marcia lab is partly funded by ITMO Cancer (18CN047-00, M.M.), Région Auvergne Rhône Alpes (project R21105CC; allocation RPH21004CCA, M.M.), FINOVI (AAP15, M.M.), Canceropole CLARA (Oncostarter, M.M.), and by the Fondation ARC pour la recherche sur le cancer (PJA-20191209284, M.M.). The Marcia lab uses the platforms of the Grenoble Instruct-ERIC center (ISBG; UAR 3518 CNRS-CEA-UGA-EMBL) within the Grenoble Partnership for Structural Biology (PSB), supported by FRISBI (ANR-10-INBS-0005-02) and GRAL, financed within the University Grenoble Alpes graduate school (Ecoles Universitaires de Recherche) CBH-EUR-GS (ANR-17-EURE-0003). M.D.V. thanks the Italian Association for Cancer Research (AIRC) for financial support (IG 23679, M.D.V.).

## Author contributions

M.M. and M.D.V. conceived and designed the work; I.S. determined all crystal structures and performed all enzymatic and biophysical assays; J.M. and G.M. performed the MD simulations; A.A. and N.B. synthesized ARN25850; C.M. provided support with ITC measurements and ITC data analysis; J.B.R. provided support with BLI measurements and BLI data analysis; P.V. performed the QM/MM simulations; A.M.C. provided support with beamline data collection; all authors interpreted the data; I.S., J.M., M.D.V., and M.M. produced the first draft of the manuscript; all authors approved the final version of the manuscript. M.M. and M.D.V. contributed equally.

## Funding

## Competing interests

The authors declare no competing interests.
