## [Peer Review File · Nature Communications]

Targeting the conserved active site of splicing machines with specific and selective small molecule modulatorsREVIEWER COMMENTS

Reviewer #1 (Remarks to the Author):

This is a nice paper that combines X-Ray, functional assays and molecular simulations. It is a follow up of previous work by the authors, and hits an important aspect on bacterial splicing. The dual mechanism of inhibition found here for intronistat B is quite nice and results might help to design better promiscuous drugs, or in the reverse others touching only one step of the reaction mechanism, opening new ways to investigate the systems.

The protocol used experimentally to dissect the effect of the drug on different stages of the process is quite nice, even the structures obtained are in some cases of low quality (not sure a 4 Å resolution structure can be used to describe many details. I have also some concerns on the theoretical calculations. However, overall, the story is convincing, but there are several points that requires attention from the authors.

The drug seems to be Mg²⁺ chelating with no specific recognition with the binding site, most of the contacts are with phosphates (only (based on the orientation in the plot) a weak H-bond with A181). The claims on specificity and relation with conserved residues are not really supported by the experiments. For example, the conserved U375 is recognized just by the phosphates, and seems far from the action site. Specificity if exist (no direct evidence is really provided) should be structure rather than sequence based. In fact, this lack of sequence specificity could explain binding promiscuity.

The authors assume the pyrogallol ring is anionic despite the pKa. I tend to agree with the authors, a negative charge will facilitate chelation. However, distances in the Figures are not so different among oxygens as expected from 1 O⁻ and 2 OH (not clear if the crystal has enough resolution to the decimals shown in the Figure), furthermore, the bromine derivative which should have a quite different pKa seems to be equally active. In summary, there are not direct evidences supporting a crucial point in the paper. I am sure the authors can run a QM/MM simulation and check which ionic state is more compatible with the electron density experimentally detected.

The paper is very difficult to follow. Figures are discussed in a no sequential ordering (example jump from Figure 1 to 5, S7 is discussed (if I am not confused) before S6. The pyrogallol ring is supposed to be anionic, but it appears neutral in the figure, without advice to the reader that its bioactive form should be anionic. The reader is force to jump from figure to the other and take a look to Marcia's previous paper to link the binding mode of the drug to the binding mode of the substrate. Example, not obvious what is the triplex the authors refer to in Figure 4 without going to 4FAW a description of the key interactions for substrate and drug will be very helpful to make the live of the reader easier (perhaps a summary cartoon?). A

Figure S5 should be complemented with specific tracing of the key contacts in the binding mode (those discussed in the X-Ray part). Figure 2D indicates a disruption of the geometry that seems to be linked to a hydroxyl rotation that leaves one Mg²⁺ uncoordinated, in apparent disagreement with experimental data. I would suggest extending the simulation (it seems transition might be reversible, but also the second hydroxyl might jump leaving Mg²⁺ uncoordinated). This type of behavior is seen in Figure S6 and again in Figure 3 (histograms are misleading when used for a non-equilibrated trajectory). The authors should extend trajectories (500 ns is not as the authors claim "extensive"). In fact, the authors claim in the discussion simulations are microsecond-long (hope this was only one more overstatement, or a typo). Furthermore, force field parameters for the drug at the binding site should be revised. The authors know well Mg²⁺ is a nightmare for classical simulations.

Figure S7 indicates a binding mode defined from a low resolution structure 4 Å (page 8). It would be nice to have MD simulations to check for maintenance of the interactions. No clear why TI simulations are done only with one of the complexed structures

The TI calculations (certainly described a little bit before 2015-2017) couple to thermodynamic cycles are nice as indicate a way to design more active compounds, even though no new compound is derived. However, it is not clear why they consider only of the binding states. It will largely enrich the paper to see the same done for the different binding modes. Technically, I would love to see also the impact of the movement of pyrogallol movement moving appart from the Mg²⁺. Is this present in all

the complexes?

There is an abuse of adjectives along all the text and an excess of "over claims" on page 15 (lines 25-27), also in page 16 15-18, or at the end of the discussion. The authors might have tuned down the impact and novelty of the results presented here.

Reviewer #2 (Remarks to the Author):

The manuscript by Silvestri et al describes the targeting, using small molecule compounds, of self-splicing group II introns. Using a combination of biochemical assays, X-ray crystallographic structures and computational approaches, an inhibition mechanism is proposed. The authors present this work as a valuable basis for the rational development of inhibitors targeting group II introns more specifically, with therapeutic prospects ranging from antifungals to cancer therapy.

The results presented are interesting, particularly as crystallographic structures featuring specific RNA/ligand complexes are very rare. These crystal structures make a significant contribution to our understanding of the mode of action of these inhibitors, and to the development of new compounds. However, there are a few problematic points:

Overall, I find that the authors tend to overestimate the contributions of this study in relation to existing knowledge. Targeting group II introns with small molecules is not a novelty by itself: for instance, use of Intronstat B was described in Fedorova et al, 2018 (this work is cited several times in the manuscript) and use of Mitoxantrone was described by Liu et al, 2021. Some sections deserve to be toned down. Here are a few examples: page 3, l8 "Here we now demonstrate experimentally that the conserved RNA-based active site of splicing ribozymes can recognize small organic compounds specifically and selectively"; page 17, l7 "In summary, our data demonstrate that the conserved RNA-based active site of splicing enzymes can specifically and selectively recognize small molecule inhibitors".

The biochemical work seems to have been carried out and interpreted correctly. The work on crystallographic structures is very substantial, but some points need attention:

- given the relatively modest resolution of the various structures (from 2.8 to 4.0 Å - and as such I don't think one can refer to high-resolution structures as on page 3 l.14-15) and the notably high Wilson B-factor from Table S2, I'm very skeptical about the discussion of the presence of metal ions and their importance in the observed complexes and mechanisms of action. It is indeed hard to believe that data acquired at this resolution and with such high temperature factors would actually allow Mg²⁺ and Na⁺ ions to be observed. Such ions are discussed along the manuscript (example: page 12, l14-15: "In line with this observation, the catalytic ions M1 and M2, which are normally released from the active site in the toggled state, are instead still present in our structure.") and visible in several figures (Fig. 2, Fig. 4, Fig. 6, Fig. S7). Given their potential importance, the presence of these ions must be assessed by data, and it seems necessary to present such data in the form of an electronic density map (possibly in stereo) in the supplementary material section.

- Several figures showing the intron active site structure are lacking clarity (Fig. 4 and Fig. 6 should be improved).

- Page 9, l.18-19: an omit map was generated before modelling the compound. I don't see the point: if the map was calculated before introducing the ligand into the model, phases are not biased by the ligand and a difference map should be used.

Other issues to be clarified:

The specific binding of compounds to the self-splicing intro was shown using biochemical assay and by Xray structures. In addition, MD simulations are presented: page 7, l22 "We further confirmed the specificity of the interaction by two independent ~500 ns-long molecular dynamics (MD) simulations...". With all due respect for molecular dynamics, and despite the fact that force fields for nucleic acids are notoriously ill-defined (Mrazikova et al J Chem Inf Model 2021; Lazzeri et al, BiophysJ 2023; Liebl & Zacharias, BiophysJ 2023), I don't think this is the method of choice for attesting to the specificity of such an interaction. On the other hand, I don't understand why robust experimental approaches (such as SPR) have not been used in this study, when the authors have the environment and equipment needed for such experiments. Obtaining precise Kd, k(on) and k(off) - or even thermodynamic parameters using calorimetric approaches - would be extremely valuable for MD developed in this study.

Minor issue: page 2 l 23-24 "...the exact comprehension of whether small molecules can specifically and/or selectively bind RNA remains quite challenging". There are now many examples of small molecules binding very specifically to RNA (for instance macrolide and aminoglycoside binding to 23S and 16S rRNA), so the question of feasibility no longer really arises.

Reviewer #3 (Remarks to the Author):

In this manuscript, Silvestri and colleagues present a series of crystal structures of a group II self-splicing intron bound in the active site by an inhibitor compound intronistat B at various stages of splicing. The authors complement their structural analysis with kinetic assays, molecular dynamics simulations and free energy calculations to explain how intronistat B inhibits both catalytic steps of splicing with different potency. The RNA-based active site of group II self-splicing introns is highly conserved in the eukaryotic spliceosome complex. Thus, the authors present intronistat B as a case study of how elucidating the mechanism of small molecule inhibitors could lead to better understanding of splicing catalysis and also inform rational design of therapeutic spliceosome inhibitors. The authors further attempt to establish a framework to comprehensively characterise RNA-binding small molecules, which could be of use to a broad scientific audience.

While we appreciate the authors' work, there are two aspects that must be addressed before publication can be considered. First, we have serious concerns regarding the authors' interpretation of the crystallographic data (see major comment 1). Second, the current manuscript text and figures lack clarity and frequently appropriate context (see major comment 2). This makes the manuscript inaccessible to non-experts outside of group II intron structural studies. We thus recommend substantial revisions.

Major comments:

1. According to the PDB validation report, the densities for intronistat B were overinterpreted. We are not convinced that there is intronistat B density in the maps of models 8OLZ and 8OM0, two of the three structures the authors present in this study. The authors should address the following concerns:

a) At the threshold displayed in the PDB validation report, the density assigned to intronistat B in the maps of models 8OLZ and 8OM0 is incomplete and fragmented, prohibiting unambiguous placement of the compound in the corresponding models. Do the authors have other crystals or better quality densities that convince of an unambiguous placement of intronistat B in these two structures? If not, we recommend the authors remove these two structures the manuscript.

b) While the quality of intronistat B density is good, the compound is slightly misplaced in model 8OLS and should be adjusted.

2. We suggest the following to improve the clarity of structural Figures 2, 4 and 6:

a) To better demonstrate the effects of the compound on the active site of the intron in the main text figures, the authors should include panels for a side-by-side comparison of their intronistat-bound structures with the available structures of the compound-free intron at an equivalent stage of splicing. This will greatly aid with delivering appropriate context of the inhibitor-bound structures.

b) The different panels in each figure should be consistent and shown from the exact same view, which does not seem to be the case in the current version of the manuscript, see the panels in Figure 2 and Figure 4 respectively. If a rotation is applied between the panels, the authors should clearly show how the panels relate to each other by indicating the direction and degrees of rotation.

c) All structural panels should have more and consistent labels (for instance, Figure 4B has no labels at all).

d) The authors should also consider to use more colour coding in all structural figures, like they did in Figure 3 panels A, B, C and F. For example, in Figure 2 and 4 to distinguish various structural elements of the group II intron active site.

Minor comments:

1. The authors quote an apparent IC₅₀ of intronistat B but do not show a dose-response curve used to derive this value. The authors should include a dose response curve of the compound from one time point in either Figure 1 or Figure S1.

2. In several places in the main text, the authors list the distances between the catalytic ions and various functionally important atoms determined from their crystal structures. These large pieces of text spanning five or more lines are disruptive to the flow of the text. Since the same information is more efficiently conveyed in the corresponding figures, we suggest the authors remove it from the main text.

3. The authors should comment on why brominated intronistat B (Figure 2C) binds to the active site of the intron in a different pose than the unmodified compound (Figure 2B).

4. To help the reader contextualise the structural figures, the authors should provide an annotated secondary structure diagram of the I1 self-splicing intron in a supplementary figure or in one of the main figures.

5. The choice of intron constructs and ionic conditions to prepare crystallographic samples of the various splicing stages should be briefly but clearly explained in the main text.

6. Some panels in Figure 3 showing the data from molecular dynamics simulations are only referred to only much later in the manuscript, which makes it hard to follow. The authors might want to consider moving individual panels from Figure 3 to a relevant structural figure, for example, panels 3C and 3D thematically belong to Figure 4, panels 3E and 3F – to Figure 6.

7. The author should explain in more detail both the rationale and the outcome of their free energy calculations.

8. It would be helpful if in Figure 7, the authors would indicate what experimental structure (by Figure number or PDB ID) each spliceosome diagram represents in their abstract schematic.

9. In their discussion, the authors should clarify the implications of pharmacological group II intron inhibition to retrotransposition.

10. On page 12, the authors refer to their second step structure of the intron as the "sodium structure". The authors should avoid using such colloquial terms not established in the field.

11. In the structural figures, the authors might want to consider showing the backbone phosphates that coordinate the catalytic metal ions in the intron active site.

12. In Figure S13, the authors could also show the same view of their structures of the compound-bound self-splicing intron. This would convey the conservation of the active site on the two splicing machineries.

RESPONSE TO EDITORIAL ADVISOR'S COMMENTS FOR:

Targeting the conserved active site of splicing machines with specific and selective small molecule modulators

Ilaria Silvestri^{1,2}, Jacopo Manigrasso¹, Alessandro Andreani¹, Nicoletta Brindani¹, **Caroline Mas³, Jean-Baptiste Reiser³, Pietro Vidossich¹, Andrew A. McCarthy²**, Marco De Vivo^{1,*}, Marco Marcia^{2,*}

*To whom correspondence should be addressed.

E-mail: marco.devivo@iit.it; mmarcia@embl.fr

We would like to thank the editor and all three reviewers for their careful assessment and overall appreciation of our manuscript and for their insightful comments and criticism. In this detailed point-by-point response to the reviewers, we address all concerns of the reviewers. We also enclose a revised version of the manuscript and figures, in which revisions are marked in red.

Responses to reviewer #1:

This is a nice paper that combines X-Ray, functional assays and molecular simulations. It is a follow up of previous work by the authors, and hits an important aspect on bacterial splicing. The dual mechanism of inhibition found here for intronistat B is quite nice and results might help to design better promiscuous drugs, or in the reverse others touching only one step of the reaction mechanism, opening new ways to investigate the systems.

The protocol used experimentally to dissect the effect of the drug on different stages of the process is quite nice, even the structures obtained are in some cases of low quality (not sure a 4 Å resolution structure can be used to describe many details. I have also some concerns on the theoretical calculations. However, overall, the story is convincing, but there are several points that requires attention from the authors.

We thank the reviewer for recognizing the value of our investigations.

The drug seems to be Mg²⁺ chelating with no specific recognition with the binding site, most of the contacts are with phosphates (only (based on the orientation in the plot) a weak H-bond with A181). The claims on specificity and relation with conserved residues are not really supported by the experiments. For example, the conserved U375 is recognized just by the phosphates, and seems far from the action site. Specificity if exist (no direct evidence is really provided) should be structure rather than sequence based. In fact, this lack of sequence specificity could explain binding promiscuity.

We thank the reviewer for this comment. We would like to clarify that we consider the binding of intronistat B to the intron's active site specific, because our X-ray crystallography data (particularly the data sets in the presence of the dibrominated intronistat B derivative) unequivocally show that intronistat B only and exclusively interacts with the intron active site. The compound does not recognize any other region of the group II intron, regardless of the splicing step being characterized. These data strongly suggest that intronistat B is recognized by specific catalytic structural features that are unique to the splice site, and not to any other RNA motif. Notably, intronistat B is selective for group II introns and does not bind other structured RNAs, such as group II intron constructs missing the catalytic domain, RNA stem-loops, or tRNAs – as previously showed by Fedorova et al. (Fedorova et al., 2018).

Indeed, our results show that the two-metal-ion (Mg^{2+}) active site of group II intron can bind intronistat B through the chelation of both catalytic ions and a few interactions with conserved nucleobases.

This said, we agree with the reviewer that the majority of the interactions between intronistat B and the intron are not sequence-specific and that this lack of sequence-specificity could explain the binding of intronistat B to both group IIB and group IIC introns.

We have modified the text to clarify these considerations on page 17, lines 8-27.

The authors assume the pyrogallol ring is anionic despite the pKa. I tend to agree with the authors, a negative charge will facilitate chelation. However, distances in the Figures are not so different among oxygens as expected from 1 O⁻ and 2 OH (not clear if the crystal has enough resolution to the decimals shown in the Figure), furthermore, the bromine derivative which should have a quite different pKa seems to be equally active. In summary, there are not direct evidences supporting a crucial point in the paper. I am sure the authors can run a QM/MM simulation and check which ionic state is more compatible with the electron density experimentally detected.

We thank the reviewer for this comment. As suggested by the reviewer, we have now additionally performed hybrid quantum mechanical / molecular mechanical (QM/MM) simulations to further investigate which ionic state is compatible with the electron density experimentally detected. Our simulations show that, when intronistat B is deprotonated, the ligand maintains the coordination to both catalytic Mg^{2+} ions for the whole simulation time of ~5.3 ps (**Figure S25a,b**). The whole coordination sphere of the two divalent metal ions is kept stable during the simulations. Consistently, the exact same behavior was observed for the deprotonated form of the brominated analog of intronistat B (**Figure S25c,d**). On the contrary, the protonated form of intronistat B lost the coordination of the Mg ions already during the preliminary equilibrium MD. We overcame this problem by introducing a soft restraint between the pyrogallol oxygens of intronistat B and the Mg ions to maintain the coordination while equilibrating the system. But despite this restraint, the subsequent QM/MM simulation confirmed that in the neutral, protonated state the ligand intronistat B is not capable of maintaining the coordination to one of the Mg ions, detaching from M1 after ~3 ps of the simulation (**Figure S25e,f**). In conclusion, the QM/MM simulations further corroborate, from a dynamical perspective, that the deprotonated form of the pyrogallol moiety is the preferred ionic state for the binding to the two-metal ions active site, in line with its thermodynamic stability estimated via pK_a calculations.

We have edited the manuscript to include these new insights on page 8, lines 27-29, page 27 lines 10-27.

The paper is very difficult to follow. Figures are discussed in a no sequential ordering (example jump from Figure 1 to 5, S7 is discussed (if I am not confused) before S6). The pyrogallol ring is supposed to be anionic, but it appears neutral in the figure, without advice to the reader that its bioactive form should be anionic. The reader is forced to jump from figure to the other and take a look to Marcia's previous paper to link the binding mode of the drug to the binding mode of the substrate.

We thank the reviewer for this comment. We have modified the text throughout the manuscript to increase clarity. We have also renumbered all the figures consecutively and in the order by which they appear in the text, to avoid confusion. We have added a note in the legend of **Figure 5** that the bioactive form of intronistat B is anionic. Finally, we have modified **Figures 2, 4 and 6** to make the comparison to our previous work clearer.

Example, not obvious what is the triplex the authors refer to in Figure 4 without going to 4FAW a description of the key interactions for substrate and drug will be very helpful to make the life of the reader easier (perhaps a summary cartoon?).

We thank the reviewer for this observation. We have modified **Figure 4** to make the comparison to structure 4FAW clearer. Our text also includes a description of the key interactions established by the inhibitors with the intron, in **Figure S6**.

A Figure S5 should be complemented with specific tracing of the key contacts in the binding mode (those discussed in the X-Ray part).

We thank the reviewer for this observation. We note that our original **Figure S5** (currently **Figure S8**) does not report atomic contacts that describe the binding mode of intronistat B, so this comment of the reviewer was unclear. We suspect the reviewer was referring to **Figure S3** (now **Figure S6**), where we do depict all atomic contacts between intronistat B and the intron active site. Thus, we have now improved **Figure S6** to accurately record all interactions that we describe in the X-ray structures.

Figure 2D indicates a disruption of the geometry that seems to be linked to a hydroxyl rotation that leaves one Mg²⁺ uncoordinated, in apparent disagreement with experimental data. I would suggest extending the simulation (it seems transition might be reversible, but also the second hydroxyl might jump leaving Mg²⁺ uncoordinated). This type of behavior is seen in Figure S6 and again in Figure 3 (histograms are misleading when used for a non-equilibrated trajectory). The authors should extend trajectories (500 ns is not as the authors claim “extensive”). In fact, the authors claim in the discussion simulations are microsecond-long (hope this was only one more overstatement, or a typo). Furthermore, force field parameters for the drug at the binding site should be revised. The authors know well Mg²⁺ is a nightmare for classical simulations.

We thank the reviewer for this comment. We have modified the text by removing the terms “extensive” and “microsecond”, on page 12, line 16 and page 16, line 8. Moreover, to improve our sampling, we have now performed two additional 500 ns-long simulation replicates of the free intron bound to intronistat B. These simulations confirmed that at the first step of splicing the binding of intronistat B is less stable (**Figure S26**), in agreement with splicing assays showing less inhibition of this splicing step. Nonetheless, the presence of intronistat B is still able to alter the dynamics of the active site and hamper the formation of the K1-N7 interaction, providing further molecular rationale to the intronistat B-driven splicing inhibition (**Figure S26**). Importantly, we have decided to increase our simulation sampling by running additional replicates in the same timescale, rather than extending the simulation time, based on our previous experience on the same system. Indeed, we have previously shown that relevant K1 dynamics can be observed within 250 ns (Manigrasso et al, 2021). Similarly, within the simulation time monitored in our current study, we show that when intronistat B is not bound to the intron, K1 interacts with N7@G288 within timescales much shorter than 500 ns. K1 interacts with N7@G288 within timescales much shorter than 500 ns, that is within 50 to 100 ns (**Figure 6G**, green and red lines). Instead, the same K1-N7@G288 interaction is not observed within the same timescale when intronistat B is bound to the RNA (**Figure 6G**, blue and yellow lines). This evidence, suggests that the simulation timescales that we have used are sufficient to describe how the presence of intronistat B prevents the dynamics of K1 and impairs the role of the ion in the splicing progression.

Furthermore, we would like to note that the less stable chelation of Mg²⁺ ions by intronistat B when bound to the exon-free intron (**Figure 6F**) is not a surprising event. Indeed, without the native substrate, the active site is intrinsically more flexible, leading to the separation of the Mg²⁺ ions, which are no longer found at 4 Å from each other – which X-ray shows to be a requirement for cations chelation by intronistat B. Nonetheless, intronistat B is still able to recognize and chelate the two metal ions and transiently stabilize them in a pseudo-catalytically competent state (4 Å from each other) in the absence of the substrate – thus before the first splicing step – even though less steadily with respect to intron as bound to the native substrate – just before the second splicing step (**Figure 6F**). Remarkably, the distorted

binding of intronistat B observed in the simulations before the first splicing step is qualitatively in line with splicing assays. Indeed, together these experiments and simulations suggest that the second step is inhibited more than the first one because the structural architecture of the intron before the second step facilitates and stabilizes the binding of the drug at the active site. Finally, we would like to note that our new QM/MM calculations confirmed the Mg-pyrogallol chelation geometries observed in the X-ray structures as well as during MD simulations, suggesting that the chosen Mg parameters were good enough in observing the dynamics of interest.

Figure S7 indicates a binding mode defined from a low-resolution structure 4 Å (page 8). It would be nice to have MD simulations to check for maintenance of the interactions. No clear why TI simulations are done only with one of the complexed structures. The TI calculations (certainly described a little bit before 2015-2017) couple to thermodynamic cycles are nice as indicate a way to design more active compounds, even though no new compound is derived. However, it is not clear why they consider only of the binding states. It will largely enrich the paper to see the same done for the different binding modes. Technically, I would love to see also the impact of the movement of pyrogallol moving apart from the Mg²⁺. Is this present in all the complexes?

We thank the reviewer for this comment. We would first of all like to reassure the reviewer that we have simulated both RNA-inhibitor complexes, covering both binding modes of intronistat B to the group II intron active site, thus addressing the reviewer's concern (**Figure 3, S8-9, and the novel Figure S26**). We have not simulated the states displayed in former Figure S7 (now **Figure S10**), because the compound is absent in those states, as explained in the figure legend and in the main text.

Additionally, as suggested by the reviewer, we have now performed new TI calculations starting from the binding complex as obtained in the absence of the substrate – i.e., before the first splicing step – to further investigate the binding mode of intronistat B within the active site of the free intron. Specifically, we have performed three replicates for the transformation compound 8 → intronistat B – the same transformation we used as a reference for the 5'exon-bound complex. In line with *i*) splicing assays, showing that the compound inhibits less this catalytic stem, *ii*) X-ray structures, showing that the compound is more water exposed in this state, and *iii*) equilibrium MD simulations, showing that the compound is more flexible in the absence of the native substrate, also TI simulations show enhanced compound flexibility of the N-tail (**Figure S15**). As such, the relative binding free energy estimated when starting from the binary complex (intron-intronistat B) complex was associated with larger errors ($\Delta\Delta G = -4.0 \pm 1.5 \text{ kcal}\cdot\text{mol}^{-1}$) and poorly converged (**Figure S14**), as compared to those estimated for the ternary complex (intron-exon-intronistat B, $\Delta\Delta G = -1.12 \pm 0.7 \text{ kcal}\cdot\text{mol}^{-1}$, **Figure S9**). In conclusion, our additional TI calculations confirm that the ternary complex can be reliably used to predict the $\Delta\Delta G_{AB}$ of compounds binding the intron active site. We have now modified the text to include these additional calculations, on page 13 lines 26-37-4.

With regard to TI calculations, we further want to clarify that we have used these calculations as a retrospective validation of the binding mode with a series of already-known compounds. Indeed, for alchemical free energy calculations to be effectively reliable, the binding mode of the ligands must be modeled very accurately to account for all the interactions responsible for the experimentally observed binding potency. Otherwise, the free energy estimates of the bound state could be dramatically affected, resulting in an inaccurate estimate of the total $\Delta\Delta G_{AB}$ (Hahn et al., 2022; Mey et al., 2020). To better clarify our approach, we have now further elaborated this concept within the method section of the current manuscript, on page on page 25, lines 13-37 and on page 26, lines 1-6

Finally, we note that during the TI transformation the flexibility of the ligand is due to dynamics of the N-tail of intronistat B, rather than to the core, which preserves its binding mode with the catalytic metals Mg²⁺ (**Figure S9 and S15**).

There is an abuse of adjectives along all the text and an excess of “over claims” on page 15 (lines 25-27), also in page 16 15-18, or at the end of the discussion. The authors might tune done the impact and novelty of the results presented here.

We thank the reviewer for this comment. To avoid any “over claims” and abuse of adjectives, we have now rephrased the text, currently located on page 18, lines 5-8 and lines 33-35.

Responses to reviewer #2:

The manuscript by Silvestri et al describes the targeting, using small molecule compounds, of self-splicing group II introns. Using a combination of biochemical assays, X-ray crystallographic structures and computational approaches, an inhibition mechanism is proposed. The authors present this work as a valuable basis for the rational development of inhibitors targeting group II introns more specifically, with therapeutic prospects ranging from antifungals to cancer therapy. The results presented are interesting, particularly as crystallographic structures featuring specific RNA/ligand complexes are very rare. These crystal structures make a significant contribution to our understanding of the mode of action of these inhibitors, and to the development of new compounds.

We would like to thank the reviewer for the appreciation of our results.

However, there are a few problematic points. Overall, I find that the authors tend to overestimate the contributions of this study in relation to existing knowledge. Targeting group II introns with small molecules is not a novelty by itself: for instance, use of Intronestat B was described in Fedorova et al, 2018 (this work is cited several times in the manuscript) and use of Mitoxantrone was described by Liu et al, 2021. Some sections deserve to be toned down. Here are a few examples: page 3, l8 “Here we now demonstrate experimentally that the conserved RNA-based active site of splicing ribozymes can recognize small organic compounds specifically and selectively”; page 17, l7 “In summary, our data demonstrate that the conserved RNA-based active site of splicing enzymes can specifically and selectively recognize small molecule inhibitors”.

We thank the reviewer for this comment. As the reviewer correctly points out it had already been shown that group II introns can be targeted with small molecules. However, none of the previous studies could prove experimentally and at the molecular level what is the mechanism of intron inhibitors. Explaining the mechanism of such inhibitors, and specifically proving experimentally that they target the intron active site, is the main contribution of our work. To highlight this specific message without overestimating the contribution of our work, we have now rephrased the text in several instances, including on page 3, lines 7-13, on page 18, lines 5-8, and on page 18, lines 33-35.

The biochemical work seems to have been carried out and interpreted correctly. The work on crystallographic structures is very substantial, but some points need attention: - given the relatively modest resolution of the various structures (from 2.8 to 4.0 Å - and as such I don't think one can refer to high-resolution structures as on page 3 l.14-15) and the notably high Wilson B-factor from Table S2, I'm very skeptical about the discussion of the presence of metal ions and their importance in the observed complexes and mechanisms of action. It is indeed hard to believe that data acquired at this resolution and with such high temperature factors would actually allow Mg²⁺ and Na⁺ ions to be observed. Such ions are discussed along the manuscript (example: page 12, l14-15: “In line with this observation, the catalytic ions M1 and M2, which are normally released from the active site in the toggled state, are instead still present in our structure.”) and visible in several figures (Fig. 2, Fig. 4, Fig. 6, Fig. S7). Given their potential importance, the presence of these ions must be assessed by data, and it seems necessary to present such data in the form of an electronic density map (possibly in stereo) in the supplementary material section.

We thank the reviewer for this comment. To address this comment, we have removed the term “high-resolution” on page 3, line 12. Furthermore, we would like to point out that our current work builds on a large body of previous crystallographic work on the same system (20 previously reported crystal structures, described in the following publications, which are cited

in our manuscript: Marcia and Pyle, Cell, 2012; Marcia and Pyle, RNA, 2014; Marcia, Methods in Mol Biol, 2016; Manigrasso et al, Nat Comm, 2021). Such previous studies had systematically confirmed the identity and characterized the structure-functional role of all intron-bound ions visible crystallographically, in the active site and beyond. We have now explicitly stated in the text that the identity of the ions had been determined in previous studies (page 7, lines 10-12).

- Several figures showing the intron active site structure are lacking clarity (Fig. 4 and Fig. 6 should be improved).

We have changed **Figures 4 and 6** to address the reviewer's comment.

- Page 9, l.18-19: an omit map was generated before modelling the compound. I don't see the point: if the map was calculated before introducing the ligand into the model, phases are not biased by the ligand and a difference map should be used.

We agree with the reviewer that our phrasing was confusing and we have thus modified it accordingly throughout the text. We now present the F_o-F_c map before modelling the compound for all structures that we report in our work.

Other issues to be clarified:

The specific binding of compounds to the self-splicing intron was shown using biochemical assay and by Xray structures. In addition, MD simulations are presented: page 7, l22 "We further confirmed the specificity of the interaction by two independent ~500 ns-long molecular dynamics (MD) simulations...". With all due respect for molecular dynamics, and despite the fact that force fields for nucleic acids are notoriously ill-defined (Mrazikova et al J Chem Inf Model 2021; Lazzeri et al, BiophysJ 2023; Liebl & Zacharias, BiophysJ 2023), I don't think this is the method of choice for attesting to the specificity of such an interaction. On the other hand, I don't understand why robust experimental approaches (such as SPR) have not been used in this study, when the authors have the environment and equipment needed for such experiments. Obtaining precise K_d , k_{on} and k_{off} - or even thermodynamic parameters using calorimetric approaches - would be extremely valuable for MD developed in this study.

We thank the reviewer for this comment. We have now performed biolayer interferometry (BLI) and isothermal titration calorimetry (ITC) experiments. Our new data show that intronistat B binds the intron with micromolar affinity ($k_{D, ITC} = 7.8 \pm 1.6 \mu M$, $k_{D, BLI} = 67.0 \pm 11.3 \mu M$, $k_{on} = 52.1 \pm 5.9 M^{-1} \cdot s^{-1}$, $k_{off} = 0.0030 \pm 0.0002 s^{-1}$) through with an enthalpy-driven reaction ($\Delta H = -10.7 \pm 0.9 kcal/mol$, $\Delta G = -7.00 \pm 0.14 kcal/mol$, $-T\Delta S = 3.69 \pm 0.73 kcal/mol$). We have reported these results in the main text on page 5, lines 14-20, and in the novel **Figures S4 and S5**. We have also added a description of the relevant methodology in the methods section of the manuscript on page 22, lines 12-37 and on page 23, lines 1-14.

Minor issue: page 2 l 23-24 "...the exact comprehension of whether small molecules can specifically and/or selectively bind RNA remains quite challenging". There are now many examples of small molecules binding very specifically to RNA (for instance macrolide and aminoglycoside binding to 23S and 16S rRNA), so the question of feasibility no longer really arises.

We thank the reviewer for this comment. We have now revised the text on page 2, lines 24-26, to highlight that structural information is indeed already available for RNA-small molecule complexes, but still very scarce. Specifically, we have revised the text as follows: "As a matter of fact, with only 841 structures of RNA-ligand complexes in the Protein Data Bank (PDB) as of January 16th, 2024, the exact comprehension of how small molecules specifically and/or selectively bind RNA remains quite challenging (Falese et al., 2021; Manigrasso et al., 2021b).

RNA-binding molecules typically tend to be unspecific and promiscuous, by interacting in a sequence-independent manner with the phosphodiester backbone of RNA or by intercalating stacked nucleobases inducing structural misfolding (Warner et al., 2018)."

Responses to reviewer #3:

In this manuscript, Silvestri and colleagues present a series of crystal structures of a group II self-splicing intron bound in the active site by an inhibitor compound intronistat B at various stages of splicing. The authors complement their structural analysis with kinetic assays, molecular dynamics simulations and free energy calculations to explain how intronistat B inhibits both catalytic steps of splicing with different potency. The RNA-based active site of group II self-splicing introns is highly conserved in the eukaryotic spliceosome complex. Thus, the authors present intronistat B as a case study of how elucidating the mechanism of small molecule inhibitors could lead to better understanding of splicing catalysis and also inform rational design of therapeutic spliceosome inhibitors. The authors further attempt to establish a framework to comprehensively characterize RNA-binding small molecules, which could be of use to a broad scientific audience. While we appreciate the authors' work, there are two aspects that must be addressed before publication can be considered.

We thank the reviewer for the appreciation of our work.

First, we have serious concerns regarding the authors' interpretation of the crystallographic data (see major comment 1). Second, the current manuscript text and figures lack clarity and frequently appropriate context (see major comment 2). This makes the manuscript inaccessible to non-experts outside of group II intron structural studies. We thus recommend substantial revisions.

Major comments:

1. According to the PDB validation report, the densities for intronistat B were overinterpreted. We are not convinced that there is intronistat B density in the maps of models 8OLZ and 8OM0, two of the three structures the authors present in this study. The authors should address the following concerns:

a) At the threshold displayed in the PDB validation report, the density assigned to intronistat B in the maps of models 8OLZ and 8OM0 is incomplete and fragmented, prohibiting unambiguous placement of the compound in the corresponding models. Do the authors have other crystals or better-quality densities that convince of an unambiguous placement of intronistat B in these two structures? If not, we recommend the authors remove these two structures the manuscript.

We thank the reviewer for this comment, which stimulated us to obtain new, and stronger evidence for the presence of intronistat B in all of our reported structures. We have now obtained 7 new crystal structures for our system, 3 of which with the di-brominated intronistat B derivative ARN25850: i) OiD1-5 co-crystallized with the 5'-exon-like oligonucleotide 5'-AUUUUAU-3' in presence of Mg^{2+} , K^+ and ARN25820 (PDB id: 8RUJ, **Figure 2C**) to validate the binding of intronistat B in presence of the 5'-exon; ii) the structure of OiD1-5 in presence of Mg^{2+} , Na^+ and ARN25850 (PDB id: 8RUK, **Figure 6C**) and iii) the structure of Oi1-5 in presence of Mg^{2+} , Li^+ and ARN25850 (PDB id: 8RUN, **Figure 6F**) to validate the compound binding to the toggled conformation. Based on our new data, and on the bromine anomalous signal, we are now in the position to unequivocally confirm the binding of the inhibitor in each one of our catalytic state.

Furthermore, we would like to point out the following additional considerations that further validate the presence of intronistat B in our structures:

1. We now report the real-space correlation coefficient (RSCC) values for the compound (0.92 for 8OLZ and 0.86 for 8OM0), which indicate a good compound fitting, as reported in Pearce et al., 2017.
2. We now report the comparison with the active site of apo crystals at similar resolution, where the compound electron density is absent (new panels A in both **Figure 2** and **Figure 4**).

b) While the quality of intronistat B density is good, the compound is slightly misplaced in model 8OLS and should be adjusted.

We thank the reviewer for this comment. We have modified the modeling of the compound to better fit the electron density (**Figure 2C** and new PDB validation reports).

2. We suggest the following to improve the clarity of structural Figures 2, 4 and 6:

a) To better demonstrate the effects of the compound on the active site of the intron in the main text figures, the authors should include panels for a side-by-side comparison of their intronistat-bound structures with the available structures of the compound-free intron at an equivalent stage of splicing. This will greatly aid with delivering appropriate context of the inhibitor-bound structures.

We thank the reviewer for helping us improve the quality of our figures. We have now modified **Figures 2, 4, and 6** by adding the active site of the apo crystals in panel A.

b) The different panels in each figure should be consistent and shown from the exact same view, which does not seem to be the case in the current version of the manuscript, see the panels in Figure 2 and Figure 4 respectively. If a rotation is applied between the panels, the authors should clearly show how the panels relate to each other by indicating the direction and degrees of rotation.

We thank the reviewer for helping us improve the quality of our figures. We have modified **Figures 2 and 4** reporting the same view in the corresponding panels.

c) All structural panels should have more and consistent labels (for instance, Figure 4B has no labels at all).

We thank the reviewer for helping us improve the quality of our figures. We have modified **Figures 2, 4, and 6** by adding labels in each panel.

d) The authors should also consider to use more color coding in all structural figures, like they did in Figure 3 panels A, B, C and F. For example, in Figure 2 and 4 to distinguish various structural elements of the group II intron active site.

We thank the reviewer for helping us improve the quality of our figures. We have modified **Figures 2, 4, and 6** by adding the same color code for the active site elements.

Minor comments:

1. The authors quote an apparent IC₅₀ of intronistat B but do not show a dose-response curve used to derive this value. The authors should include a dose response curve of the compound from one time point in either Figure 1 or Figure S1.

We thank the reviewer for this comment. We have added the IC₅₀ plot in the novel panel D of **Figure 1**.

2. In several places in the main text, the authors list the distances between the catalytic ions and various functionally important atoms determined from their crystal structures. These large pieces of text spanning five or more lines are disruptive to the flow of the text. Since the same information is more efficiently conveyed in the corresponding figures, we suggest the authors remove it from the main text.

We thank the reviewer for this comment. We have modified the text by removing the disruptive phrases, for instance on page 7, lines 23-24 and on page 11, line 18.

3. The authors should comment on why brominated intronistat B (Figure 2C) binds to the active site of the intron in a different pose than the unmodified compound (Figure 2B).

We thank the reviewer for this comment, which we have by adding a sentence in the main text, on page 8, lines 10-17. Briefly, we note that the pyrogallol moieties of both intronistat B and its dibrominated analogue bind the intron's active site with the same pose (**Figure 2C and D**). However, as the reviewer correctly points out, the benzofuran bicyclic group and the tail of the two compounds adopt different poses. This effect is likely due to the fact that the benzofuran ring is rotated $\sim 90^\circ$ in the brominated analogue, due to steric effects induced by the bromine atoms themselves.

4. To help the reader contextualize the structural figures, the authors should provide an annotated secondary structure diagram of the I1 self-splicing intron in a supplementary figure or in one of the main figures.

We thank the reviewer for this comment. We have now added the novel **Figure S1**, which reports the annotated secondary structure diagram of the I1 self-splicing intron from *O. iheyensis*, as suggested by the reviewer.

5. The choice of intron constructs and ionic conditions to prepare crystallographic samples of the various splicing stages should be briefly but clearly explained in the main text.

We thank the reviewer for this comment. As suggested by the reviewer, we now describe the effects of the ionic conditions during the crystallization in the main text on page 11, lines 6-7, and page 15, lines 8-11.

6. Some panels in Figure 3 showing the data from molecular dynamics simulations are only referred to only much later in the manuscript, which makes it hard to follow. The authors might want to consider moving individual panels from Figure 3 to a relevant structural figure, for example, panels 3C and 3D thematically belong to Figure 4, panels 3E and 3F – to Figure 6.

We thank the reviewer for this comment. We have modified **Figures 3, 4, and 6** according to the reviewer's suggestion.

7. The author should explain in more detail both the rationale and the outcome of their free energy calculations.

We thank the reviewer for raising this comment. We have now highlighted the rationale of our free energy calculations in the methods section (page 25, lines 13-37 and page 26, lines 1-6) and clarified the outcome in the main text (page 12, lines 26-37 and page 13, lines 1-2).

8. It would be helpful if in Figure 7, the authors would indicate what experimental structure (by Figure number or PDB ID) each spliceosome diagram represents in their abstract schematic.

We thank the reviewer for this comment. We have now modified **Figure 7** by incorporating the PDB ids of all relevant structures for each state, according to the reviewer's suggestion.

9. In their discussion, the authors should clarify the implications of pharmacological group II intron inhibition to retrotransposition.

We thank the reviewer for this comment. We discuss this point in the Discussion section of our manuscript on page 18, lines 21-32.

10. On page 12, the authors refer to their second step structure of the intron as the “sodium structure”. The authors should avoid using such colloquial terms not established in the field.

We thank the reviewer for highlighting this incorrect terminology. We have corrected the text on page 13, line 24 and on page 15, lines 19 and 31.

11. In the structural figures, the authors might want to consider showing the backbone phosphates that coordinate the catalytic metal ions in the intron active site.

We thank the reviewer for this comment. We have tested various graphical outputs of our figures and believe that adding the backbone phosphates in stick representations would make the figures too crowded and unclear. We have therefore instead changed the graphics of **Figures 2, 4, and 6** to make the figures clearer. We have also used a different color code to indicate distances of coordination bonds vs distances of hydrogen bonds.

12. In Figure S13, the authors could also show the same view of their structures of the compound-bound self-splicing intron. This would convey the conservation of the active site on the two splicing machineries.

We thank the reviewer for this comment. We have now modified former **Figure S13** (now **Figure S16**) according to the reviewer’s suggestion.

REVIEWER COMMENTS

Reviewer #1 (Remarks to the Author):

I think the authors have made a good effort to solve all questions and criticisms provided by me and the other reviewers

Reviewer #2 (Remarks to the Author):

The revised version of the manuscript "Targeting the conserved active site of splicing machines with specific and selective small molecule modulators" was significantly improved. The authors have made significant efforts to address the majority of the referees' comments. However, there are still two points I would like to come back to.

The first one is about interactions studies now presented page 5 and in Fig S4 and S5. I assume that the authors will agree that BLI fits Fig S5 are not really acceptable, and that this may be due to the approach, which might be not suitable for this study. I think it's perfectly possible to be satisfied with the ITC data obtained. According to figure S4, complete saturation with the ligand is not obtained, but I understand that this may be due to solubility problems of the compound (ideally, given the low Wiseman coefficient of this experiment, it would be necessary to reach a molar ratio of 3-5, instead of 2). All that's missing from this ITC experiment is a blank control by injecting the compound into the buffer. The signal generated must be taken into account to obtain a correct value for the dilution heat correction (and more accurate DH, Kd), especially since the buffer contains 1% DMSO. Taking this blank into account will enable a more correct fit to be achieved than fixing the stoichiometry at 1.0 as noted in the materials and methods (page 22, line 18). Indeed, if the compound is not completely soluble, the wrong estimation of a concentration will have dramatic consequences on all thermodynamic parameters (thus including Kd and DH). Ideally, it is also always preferable to have a negative control for a ligand of this order of magnitude (μM and not nM), for example using a point mutant on the active site of the RNA. If it's available, I think it's worth doing this control to check that affinity is dramatically affected.

The second point is probably more conflictual and is about metal ions. I maintain that it is not possible to obtain reliable ion data, given the resolutions obtained, but also the unusually high average B-factors (Table S2). The publication of previous articles does not guarantee the accuracy of the claims made within them, especially considering that these earlier studies have also been reported with similar medium-range resolutions. As a consequence, I request the addition of a stereo figure showing the density map as a supplementary data so that readers can properly assess the accuracy of the data concerning these ions. Alternatively, authors should provide a clear explanation related to how they placed the ions, and what are the limitations of such a process in their manuscript and especially the limitations related.

Reviewer #3 (Remarks to the Author):

We appreciate the authors great efforts to improve the manuscript, by providing additional analyses, data, and redrawing figures. We noticed a few points of concern in this revised version, which we would ask the authors to address prior to publication.

1. The panels in Figures 4H and 6G appear to be duplicates of each other. The original version of the manuscript had two different plots, which likely got mixed up in the revised version.
2. The Ligand B factor reported in Table S2 for the structure with PDB code 8RUK reads 2230. This is most likely an error that should be corrected.

3. Owing to the low resolution of the densities containing inhibitors, we request that for the main text figures, which show inhibitors, the authors provide corresponding views and details of these structures in the supplement, but coloured by B factor. This would enable for a better assessment of model quality in the critical regions.

3. The authors should clarify why two different introns were used for different parts of the study.

4. The authors should draw and color code both introns in Fig S1. According to the color code used for the various intron RNA elements in the main text figures.

RESPONSE TO REVIEWER'S COMMENTS FOR:

Targeting the conserved active site of splicing machines with specific and selective small molecule modulators

Ilaria Silvestri^{1,2}, Jacopo Manigrasso¹, Alessandro Andreani¹, Nicoletta Brindani¹, Caroline Mas³, Jean-Baptiste Reiser³, Pietro Vidossich¹, Andrew A. McCarthy², Marco De Vivo^{1,*}, Marco Marcia^{2,*}

*To whom correspondence should be addressed.

E-mail: marco.devivo@iit.it; mmarcia@embl.fr

We would like to thank the editor and all three reviewers for their careful assessment and overall appreciation of our revised manuscript and for their further comments. In this detailed point-by-point response to the reviewers, we address all concerns of the reviewers. We also enclose a revised version of the manuscript and figures, in which new revisions are marked in red.

Responses to reviewer #1:

I think the authors have made a good effort to solve all questions and criticisms provided by me and the other reviewers.

We thank the reviewer for approving our revisions.

Responses to reviewer #2:

The revised version of the manuscript "Targeting the conserved active site of splicing machines with specific and selective small molecule modulators" was significantly improved. The authors have made significant efforts to address the majority of the referees' comments.

We thank the reviewer for appreciating the improvements in our manuscript.

However, there are still two points I would like to come back to. The first one is about interactions studies now presented page 5 and in Fig S4 and S5. I assume that the authors will agree that BLI fits Fig S5 are not really acceptable, and that this may be due to the approach, which might be not suitable for this study.

To address this comment of the reviewer, we have now improved the fitting of the BLI data, by using a biphasic 2:1 heterogenous kinetic model instead of the conventional monophasic 1:1 Langmuir model, which we had used in the first round of revision. The new fits are now represented in the updated **Figure S5**. Considering these improved fits, we would like to maintain the BLI results in the manuscript, and we would like to briefly contextualize our experiment and motivate our choice.

Initially, the reviewer had suggested us to perform SPR experiment to measure the binding kinetics between intronistat B and the group II intron. For our system, composed of a large highly structured RNA macromolecule (the group II intron) and a small organic molecule (intronistat B) such an approach is intrinsically very challenging. Binding of intronistat B to the intron immobilized on the SPR sensorchip induces small changes in refractive index and thus

in the bound masses, which are difficult to measure. The measurement is also more prone to unspecific binding of small compounds, which hinders reliable kinetic analysis. The effects of DMSO on the dextran support of the sensorchip, although they can be corrected, further increase the background signal, making it even more difficult to detect differences between the reference trace and the sample trace.

Considering these objective limitations of SPR, the price of the sensorchips, and the enzymatic and crystallographic data contained in our manuscript, we were hesitant to perform the SPR analysis. Indeed, our enzymatic data (experimental IC_{50} and K_i measurements) and our 13 crystal structures, describe at the molecular level the exact binding mode of intronistat B at the splice site, in a much more powerful way than through any other biophysical approach.

Nonetheless, to satisfy the reviewer's request, we have attempted to perform the SPR experiment. We have rigorously screened various experimental strategies and protocols. Not unexpectedly, though, we could not prevent intronistat B from binding non-specifically onto the SPR sensorchips and we thus had to abandon SPR as an approach to perform our kinetic analysis.

In parallel tests, we were however able to identify experimental conditions where intronistat B showed significantly reduced unspecific binding to BLI biosensors. Although the binding signal levels of the compound remains weak, as expected for our small molecule-RNA system, we succeeded in measuring concentration-dependent signals by BLI, indicative of binding events. This success encouraged us to complete the experiment and present the data in the manuscript. We acknowledge the significant challenges in performing such experiments and in rationalizing the complicated kinetics behavior of our biological system, but we believe that reporting our results is the most transparent way to describe our efforts, and also the most instructive approach for other colleagues that will certainly study similar RNA-small molecule systems in the future.

I think it's perfectly possible to be satisfied with the ITC data obtained. According to figure S4, complete saturation with the ligand is not obtained, but I understand that this may be due to solubility problems of the compound (ideally, given the low Wiseman coefficient of this experiment, it would be necessary to reach a molar ratio of 3-5, instead of 2). All that's missing from this ITC experiment is a blank control by injecting the compound into the buffer. The signal generated must be taken into account to obtain a correct value for the dilution heat correction (and more accurate DH , K_d), especially since the buffer contains 1% DMSO. Taking this blank into account will enable a more correct fit to be achieved than fixing the stoichiometry at 1.0 as noted in the materials and methods (page 22, line 18). Indeed, if the compound is not completely soluble, the wrong estimation of a concentration will have dramatic consequences on all thermodynamic parameters (thus including K_d and DH). Ideally, it is also always preferable to have a negative control for a ligand of this order of magnitude (μM and not nM), for example using a point mutant on the active site of the RNA. If it's available, I think it's worth doing this control to check that affinity is dramatically affected.

To address this comment of the reviewer, we have now subtracted the blank control experiment (intronistat B injected into the calorimetric cell containing only the reaction buffer) from our titration curves. Since the blank control signal is homogeneous for each injection, the thermodynamic parameters (K_d and ΔH) remain essentially unchanged.

Unfortunately, we do not have point mutants of the RNA that would constitute relevant controls for this experiment, and further ITC studies would go beyond the scope of our manuscript.

We would like to point out that, similarly to SPR/BLI discussed above, also ITC is a non-trivial and challenging technique to apply to our RNA-small molecule system. ITC requires the use of high concentrations of a compound that is poorly soluble in the reaction buffer, and of high concentrations of a very large RNA, for which we crucially need to ensure folding even in the presence of non-negligible concentrations of DMSO. The reviewer initially requested us to use ITC to derive thermodynamic parameters to describe the binding of intronistat B to the group

II intron. Although such parameters could be powerfully derived from our original enzymatic and crystallographic data, we thought that confirming them with an orthogonal biophysical technique such as ITC could provide added value to our manuscript. Accordingly, we have performed the ITC experiment and we report the respective thermodynamic results in our revised manuscript.

But the unequivocal demonstration that the compound binds at the intron active site is not offered by the ITC data, but rather by our 13 crystal structures, crucially corroborated by the anomalous scattering crystallographic data obtained for the di-brominated intronistat B analogue. As we explain in our manuscript, the crystal structures show that intronistat B primarily establishes sequence-unspecific contacts with the group II intron. Therefore, there is no rational ground to support the design of correctly folded group II intron point mutants that do not bind intronistat B anymore. Designing mutants that disrupt the intron active site structure would be irrelevant and provide no added value to our investigation.

The second point is probably more conflictual and is about metal ions. I maintain that it is not possible to obtain reliable ion data, given the resolutions obtained, but also the unusually high average B-factors (Table S2). The publication of previous articles does not guarantee the accuracy of the claims made within them, especially considering that these earlier studies have also been reported with similar medium-range resolutions. As a consequence, I request the addition of a stereo figure showing the density map as a supplementary data so that readers can properly assess the accuracy of the data concerning these ions. Alternatively, authors should provide a clear explanation related to how they placed the ions, and what are the limitations of such a process in their manuscript and especially the limitations related.

To address this comment of the reviewer, we have now produced the novel **Figure S28**, where we represent the stereo views of the $2F_o-F_c$ and the F_o-F_c omit electron density maps for the metal ions located at the intron active site.

We would like to further point out that the identification and assignment of the ions in the group II intron active site – and throughout the entire molecule – is well-established in the field. The accuracy in modeling these ions is guaranteed not by subjective claims in previous publications, but by the objective and in most cases openly accessible data underlying such publications.

First, the presence of magnesium and potassium ions in the intron active site is corroborated by several decades of biochemical and enzymatic data (for instance: Peebles et al, Cell, 1986; Steitz and Steitz, PNAS, 1993; Padgett et al, Science, 1994; Podar et al, Mol Cell Biol, 1995; Daniels et al, JMB, 1996; Ho Faix, 1998; de Lencastre et al, NSMB, 2005; Su et al, NAR, 2005; Fedorova et al, JMB, 2007).

Second, a systematic comparative crystallographic study, which importantly used metal ion replacement and anomalous scattering data coupled to functional studies to overcome the limitations posed by the resolution of the structures, allowed for the precise identification and assignment of all ions in the entire group II intron structure, not just the active site (Marcia and Pyle, Cell, 2012; Marcia and Pyle, RNA, 2014). These data – including all coordinate and structure factor files – are deposited in the PDB, and are openly accessible with PDB codes 4FAQ, 4FAR, 4FAU, 4E8K, 4E8M, 4E8N, 4E8P, 4E8Q, 4E8R, 4E8T, 4E8V, 4FAW, and 4FAX. Third, all studies mentioned above have been corroborated by all most recent crystal and cryo-electron microscopy structures of homologous group II introns (Robart et al, Nature, 2014; Chan et al, Nat Comm, 2018; Haack et al, Cell, 2019; Xu et al, Nature, 2023; Haack et al, NSMB, 2024) and even the evolutionary descendant of the intron, namely the human spliceosome (Wilkinson et al, Mol Cell, 2021; Bai et al, Science, 2021)!

In light of these considerations, we believe that further characterization of the metal ions in our structures goes beyond the scope of our manuscript.

Responses to reviewer #3:

We appreciate the authors great efforts to improve the manuscript, by providing additional analyses, data, and redrawing figures.

We thank the reviewer for appreciating the improvements in our manuscript.

We noticed a few points of concern in this revised version, which we would ask the authors to address prior to publication.

1. The panels in Figures 4H and 6G appear to be duplicates of each other. The original version of the manuscript had two different plots, which likely got mixed up in the revised version.

We thank the reviewer for pointing out this discrepancy in the figures. The corrected graphs are now shown in panel H of **Figure 4** and panel G of **Figure 6**.

2. The Ligand B factor reported in Table S2 for the structure with PDB code 8RUK reads 2230. This is most likely an error that should be corrected.

We thank the reviewer for pointing out this typo. The correct value of the overall ligand B-factor for structure 8RUK is now reported in **Table S2**.

3. Owing to the low resolution of the densities containing inhibitors, we request that for the main text figures, which show inhibitors, the authors provide corresponding views and details of these structures in the supplement, but coloured by B factor. This would enable for a better assessment of model quality in the critical regions.

To address this comment of the reviewer, we have now added **Figure S27** in which we report a representation of intronistat B in our structures, color-coded by the B-factor value of each atom.

3. The authors should clarify why two different introns were used for different parts of the study.

To address this comment of the reviewer, we explain in the main text that we have used the yeast mitochondrial ai5 γ group IIB intron of *S. cerevisiae*, as a reference for IC₅₀ calculations to compare our data with previous published literature (explained on page 4, lines 4-5 and on page 5, lines 3-6), and the bacterial group IIC intron of *O. ihoyensis*, for all further enzymatic, biochemical and biophysical studies, because this intron, differently from the yeast homologue, can be crystallized (explained on page 4, lines 10-13). We have now added a more detailed explanation of the constructs we use in the Material and Methods section of the manuscript, on page 21, lines 3-8.

4. The authors should draw and color code both introns in Fig S1. According to the color code used for the various intron RNA elements in the main text figures.

To address this comment of the reviewer, we have updated **Figure S1** by adding the secondary structure map of the ai5 γ group IIB intron of *S. cerevisiae* in panel B. We have also recolored the main elements of the active site in both introns following the same color code used in the figures of the main text.

REVIEWERS' COMMENTS

Reviewer #2 (Remarks to the Author):

The authors have made efforts to significantly improve the manuscript once again; however, some issues persist nonetheless.

First, I'd like to make some comments about SPR/BLI.

Molecular weights of the intron and intronistat B are about 130 kDa and 410 Da, respectively (ratio=317). It's well established that modern SPR instruments have very few practical limits on the molar mass of the ligands tested. Among the thousands of examples, I'll take just two: the HIV-1 reverse transcriptase protein is 117 kDa (similar to the intron) interacting with very hydrophobic NNRTI inhibitors necessitating the use of DMSO for SPR measurements, akin to Intronistat B. For example, the NNRTI Nevirapine is 266 Da, so the ratio HIV RT/Nevirapine is about 440, an even worst situation for SPR data acquisition. Many laboratories have nevertheless measured interactions between these compounds by SPR (see Geitmann et al, J Med Chem 2011).

Even more challenging scenarios arise, such as with ratios exceeding 1000, as observed in the interaction between the PFV intasome (> 400 kDa) and integrase inhibitors like Dolutegravir (419 Da). These interactions also require the presence of 5% DMSO due to solubility issues of the drug (refer to Li, Passos, Shan et al., Sci Adv 2023 9(29) DOI: 10.1126/sciadv.adg5953).

So, in the current scenario, I believe it presents some challenges, yet it's evidently not excessively intricate for SPR analysis.

Having said that, BLI measurements such as those in this case can perfectly well also answer the questions posed. It is nevertheless surprising that a multi-site model has to be applied in order to fit the data, whereas the ITC data have been fitted with a simple single-site model. Is there an explanation?

Regarding the ITC data, the authors have perfectly answered my questions.

Second, I'm once again a little cautious about the answers concerning ions in the structures presented. I consider that the data presented by the authors are barely acceptable, and the arguments cannot be considered as rock solid. I agree that Mg-O distances shown in Fig 2B are fitting with Mg distances and we can see some density around Mg in the new FigS28 presented in the supplementary section, although it's quite odd to have to go down to 0.8 sigma to observe the density around the ions (more or less around one of the K⁺ which is nevertheless an ion possessing significantly more electrons, which confirms that the data are rather weak).

In the panel B, the legend should be corrected: either the structure was calculated before modelling the ions and the resulting map is not an omit map (but a difference map), or (more likely), the structure was calculated including ions and an omit map (what kind of omit map? simple omit map, composite omit map, SA composite omit map? Polder omit map?) was used to reduce model bias regarding ions.

RESPONSE TO REVIEWER'S COMMENTS FOR:

Targeting the conserved active site of splicing machines with specific and selective small molecule modulators

Ilaria Silvestri^{1,2,†}, Jacopo Manigrasso^{1, ‡}, Alessandro Andreani¹, Nicoletta Brindani¹, Caroline Mas³, Jean-Baptiste Reiser⁴, Pietro Vidossich¹, Gianfranco Martino¹, Andrew A. McCarthy², Marco De Vivo^{1,*}, Marco Marcia^{2,*}

*To whom correspondence should be addressed.

E-mail: marco.devivo@iit.it; mmarcia@embl.fr

We would like to thank the editor and the reviewer for their careful assessment and overall appreciation of our revised manuscript and for their further comments. In this detailed point-by-point response to the reviewer, we address their remaining concerns. We also enclose a revised version of the manuscript and figures, formatted according to the editorial requests for publication of the manuscript.

Responses to reviewer #2:

The authors have made efforts to significantly improve the manuscript once again; however, some issues persist nonetheless.

First, I'd like to make some comments about SPR/BLI.

Molecular weights of the intron and intronistat B are about 130 kDa and 410 Da, respectively (ratio=317). It's well established that modern SPR instruments have very few practical limits on the molar mass of the ligands tested. Among the thousands of examples, I'll take just two: the HIV-1 reverse transcriptase protein is 117 kDa (similar to the intron) interacting with very hydrophobic NNRTI inhibitors necessitating the use of DMSO for SPR measurements, akin to Intronistat B. For example, the NNRTI Nevirapine is 266 Da, so the ratio HIV RT/Nevirapine is about 440, an even worst situation for SPR data acquisition. Many laboratories have nevertheless measured interactions between these compounds by SPR (see Geitmann et al, J Med Chem 2011). Even more challenging scenarios arise, such as with ratios exceeding 1000, as observed in the interaction between the PFV intasome (> 400 kDa) and integrase inhibitors like Dolutegravir (419 Da). These interactions also require the presence of 5% DMSO due to solubility issues of the drug (refer to Li, Passos, Shan et al., Sci Adv 2023 9(29) DOI: 10.1126/sciadv.adg5953).

So, in the current scenario, I believe it presents some challenges, yet it's evidently not excessively intricate for SPR analysis.

Having said that, BLI measurements such as those in this case can perfectly well also answer the questions posed. It is nevertheless surprising that a multi-site model has to be applied in order to fit the data, whereas the ITC data have been fitted with a simple single-site model. Is there an explanation?

Regarding the ITC data, the authors have perfectly answered my questions.

We thank the reviewer for appreciating the improvements in our manuscript.

We suspect that we had to use a biphasic Langmuir heterogeneous model to fit the BLI data to account for the heterogeneity of the group II intron adsorbed to the surface of the BLI sensorchips. Possibly, the fact that the biotinylated intron needs to be denatured and refolded before the assay may explain at least in part such heterogeneity. Indeed, biphasic behavior are also observed in splicing kinetics experiments, where the radiolabeled intron is also refolded before the assays. For ITC, instead, the intron can be purified through a non-denaturing purification method – the same used for crystallization – and this procedure likely

ensures higher homogeneity.

Second, I'm once again a little cautious about the answers concerning ions in the structures presented. I consider that the data presented by the authors are barely acceptable, and the arguments cannot be considered as rock solid. I agree that Mg-O distances shown in Fig 2B are fitting with Mg distances and we can see some density around Mg in the new FigS28 presented in the supplementary section, although it's quite odd to have to go down to 0.8 sigma to observe the density around the ions (more or less around one of the K⁺ which is nevertheless an ion possessing significantly more electrons, which confirms that the data are rather weak).

In the panel B, the legend should be corrected: either the structure was calculated before modelling the ions and the resulting map is not an omit map (but a difference map), or (more likely), the structure was calculated including ions and an omit map (what kind of omit map? simple omit map, composite omit map, SA composite omit map? Polder omit map?) was used to reduce model bias regarding ions.

We thank the reviewer for these further comments on the ion assignments. In agreement with the editor we have now modified the text in page 6, lines 34-37 and in page 7, lines 1-4. We have added the following sentence:

“While the electron density around the metal cluster in the active site is weak, the interatomic distances and coordination spheres were compatible with magnesium ions at the M1 and M2 positions and with potassium ions at the K1 position (Figure 2B and Figure S28). Importantly, we have modeled magnesium and potassium ions at the M1/M2 and at the K1 positions, respectively, because the identity of the ions that occupy these sites had been previously established based on metal-ion replacement and crystallographic anomalous scattering studies [(Marcia and Pyle, 2012, 2014a), see also a stereo view of the electron density of the active site ions in Figure S28].”

We would also like to clarify that, while we have contoured the ion $2F_o-F_c$ electron density map at 0.8 sigma in Fig S28A, the electron density is visible up to 4.0σ for M1, up to 4.5σ for M2, up to 1.6σ for K1, and up to 3.0σ for K2.

Finally, to address the reviewer's comment, we have modified the legend of Fig S28B, and clarified that the figure shows the F_o-F_c difference map obtained after the first round of refinement, when only the ribonucleic chain of the intron – and no ions, water molecules, or ligands – are present in the model.